# Spatial cycles mediated by UNC119 solubilisation maintain Src family kinases plasma membrane localisation

Antonios D. Konitsiotis[1], Lisaweta Roßmannek[1], Angel Stanoev[1], Malte Schmick[1] & Philippe I.H. Bastiaens[1,2]

The peripheral membrane proto-oncogene Src family protein tyrosine kinases relay growth factor signals to the cytoplasm of mammalian cells. We unravel the spatial cycles of solubilisation, trapping on perinuclear membrane compartments and vesicular transport that counter entropic equilibration to endomembranes for maintaining the enrichment and activity of Src family protein tyrosine kinases at the plasma membrane. The solubilising factor UNC119 sequesters myristoylated Src family protein tyrosine kinases from the cytoplasm, enhancing their diffusion to effectively release Src family protein tyrosine kinases on the recycling endosome by localised Arl2/3 activity. Src is then trapped on the recycling endosome via electrostatic interactions, whereas Fyn is quickly released to be kinetically trapped on the Golgi by palmitoyl acyl-transferase activity. Vesicular trafficking from these compartments restores enrichment of the Src family protein tyrosine kinases to the plasma membrane. Interference with these spatial cycles by UNC119 knockdown disrupts Src family protein tyrosine kinase localisation and signalling activity, indicating that UNC119 could be a drug target to affect oncogenic Src family protein tyrosine kinase signalling.

---

[1] Department of Systemic Cell Biology, Max Planck Institute of Molecular Physiology, Dortmund 44227, Germany. [2] Faculty of Chemistry and Chemical Biology, Technical University of Dortmund, Dortmund 44227, Germany. Correspondence and requests for materials should be addressed to P.I.H.B. (email: philippe.bastiaens@mpi-dortmund.mpg.de)

Activated receptors at the plasma membrane (PM) transmit signals to the cytoplasm via peripheral membrane signalling proteins[1], consisting mostly of GTPases and kinases. A major class of peripheral membrane kinases is the lipidated Src family protein tyrosine kinases (SFKs) that amplify and sustain signalling from activated receptors[2–4], which explains their oncogenic potential[4, 5]. This large family of nonreceptor-type protein tyrosine kinases consists of eight structurally related members, with c-Src (Src), Fyn and c-Yes being ubiquitously expressed[6]. All SFKs contain an N-terminally myristoylated segment, followed by Src-homology (SH) domains SH3 and SH2, the enzymatic tyrosine kinase domain and a short C-terminal tail[7]. Canonical actuation of SFK activity occurs by dephosphorylation of Y530 by protein tyrosine phosphatases that disrupts the inhibitory intramolecular interaction with the SH2 domain[8]. SFKs then fully activate by the autocatalytic phosphorylation of Y419 in the kinase activation loop[9].

The concentration of SFKs at the cytoplasmic face of the PM drives their interaction and thereby the autocatalytic phosphorylation on Y419, as well as the propagation of signals by promoting the interaction with specific effectors[6, 9]. The level of SFK enrichment at the PM is therefore a critical parameter for its activity that can be altered to affect aberrant oncogenic signalling. All SFKs can interact with membranes by co-translational attachment of a myristoyl fatty acid moiety (C14:0) to the N-terminal Gly residue, via an amide bond[10]. For most SFKs, the additional attachment of palmitic acid (C16:0) by the DHHC family of PATs to one or two N-terminal Cys residues further enhances the affinity for membranes[11]. Src is not modified with a second lipid but instead contains a stretch of basic amino acids at the N terminus that promotes interaction with negatively charged phospholipids that are abundant at the PM[12]. However, neither lipidation nor the polybasic stretch or a combination thereof is sufficient to ensure a specific localisation to the PM. This is due to endocytic processes and spontaneous dissociation that enable the exchange of SFKs among membranes, leading to their thermodynamic equilibrium to extensive endomembrane surfaces[13–15]. As demonstrated for farnesylated Ras proteins, cells use energy-driven spatial cycles to counter this entropy-driven equilibrium to endomembranes[15, 16]. The GDI-like solubilising factor (GSF), PDEδ (delta subunit of phosphodiesterase-6) has an essential role in these cycles by sequestering-dissociated Ras from membranes via its farnesyl moiety to promote diffusional exploration of the cell[17]. Localised release of Ras from PDEδ by the allosteric interaction with the GTPase Arl2 in a GTP-dependent manner then concentrates it on perinuclear membranes[14, 18]. From there, electrostatic interaction traps KRas on the recycling endosome (RE), whereas palmitoylation traps palmitoylatable H-Ras and N-Ras on Golgi membranes. From these organelles, Ras proteins are shuttled back to the PM by vesicular transport to maintain an out-of-equilibrium concentration at the PM[15].

The UNC119A and UNC119B proteins are structurally related to PDEδ, containing a hydrophobic pocket that, unlike PDEδ, can sequester myristoylated and lauroylated proteins[19]. This interaction is critical for the trafficking of myristoylated proteins to the cilium of the cells[19], and structural studies indicate that the interaction with the GTPases Arl2 or Arl3 allosterically releases myristoylated cargo[20, 21]. It has also been shown that UNC119 activates the SFKs Lyn, Lck and Fyn in haemopoietic cells and is important for the formation of the immunological synapse in T cells[22, 23].

Here we show that UNC119 is the GSF for myristoylated SFKs that mediate spatial cycles to maintain their localisation on the PM. The spatial organisation of the Arl-mediated release of SFKs from UNC119 is analogous to that of Ras protein release from PDEδ, revealing a conserved mechanism to maintain the localisation of lipidated peripheral membrane molecules.

## Results

**Vesicular transport maintains Src and Fyn PM localisation.** In order to compare the localisations of polybasic-stretch-containing Src and dually palmitoylated Fyn, both SFKs were C-terminally fused to mCitrine (Src-mCit and Fyn-mCit) and expressed in HeLa cells. These recombinant proteins preserved the regulation of their autocatalytic activity and were expressed at approximately equal or lower levels as compared to endogenous SFKs (Supplementary Fig. 1).

Both SFKs are enriched at the PM, as demonstrated by their co-localisation with the PM marker, mCherry-tk-Ras (Fig. 1a). In addition, Src-mCit exhibited enrichment on perinuclear membranes, while Fyn-mCit was barely visible on internal membranes. Immunofluorescence (IF) experiments with Rab11a-specific antibodies (Fig. 1a and Supplementary Fig. 2a) demonstrated that Src-mCit co-localised with the Rab11a-positive RE compartment, while no co-localisation was apparent for Fyn-mCit (Fig. 1a). In order to investigate whether vesicular recycling through the RE is relevant for Src PM localisation, we co-expressed Src-mCit with a fluorescent dominant-negative mutant form of Rab11 (Rab11$^{S25N}$-BFP (blue fluorescent protein)), which inhibits recycling of proteins from the RE[24]. Live imaging of Src-mCit or Fyn-mCit in HeLa cells expressing Rab11$^{S25N}$-BFP showed the accumulation of Src-mCit in large vesicular structures containing Rab11$^{S25N}$-BFP in the perinuclear region, while the localisation of Fyn-mCit was unaffected (Fig. 1b and Supplementary Fig. 2b). Knockdown (KD) of Rab11 by short interfering RNA (siRNA) resulted in a similar perinuclear accumulation of Src-mCit in large vesicular structures, whereas Fyn-mCit localisation was unaffected (Fig. 1c, d)[25]. These experiments show that vesicular recycling via the RE is essential for the localisation of Src on the PM. Despite Fyn-mCit exhibiting a more pronounced PM localisation as compared to Src-mCit, we could detect dim fluorescence on perinuclear structures that were distinct from the Rab11a-positive RE in 51% of the cells ($n = 217$; Fig. 1a). We examined whether these structures corresponded to the Golgi apparatus since Fyn-mCit is dually palmitoylated on two C-terminal cysteines and palmitoylation of N/HRas has been shown to predominantly occur on the Golgi[13]. Live imaging of HeLa cells at 37 °C, co-expressing Fyn-mCit and the *trans*-Golgi protein β-1,4-galactosyltransferase (GalT) C-terminally tagged with mCerulean (GalT-mCer), showed low co-localisation (Fig. 1e, *left panels*)[13, 26]. To test whether the low amount of Fyn-mCit at the Golgi was due to low palmitate turnover kinetics, we slowed anterograde secretory transport by applying a temperature block (20 °C for 24 h). This resulted in a shift in the distribution of Fyn-mCit but not in Src-mCit towards the Golgi (Fig. 1e, *right panels*). This was not affected by long-term incubation with the protein synthesis inhibitor, cyclohexamide, demonstrating that the temperature block induced a shift in the steady-state distribution of Fyn-mCit from predominantly PM to the Golgi (Supplementary Fig. 2c). This is consistent with a palmitoylation cycle, analogous to that of N/HRas, which maintains a steady-state distribution of Fyn at the PM[13]. To further substantiate this hypothesis, we investigated the effect of acyl-protein thioesterase (APT) inhibition on the Golgi localisation of Fyn-mCit. APTs depalmitoylate peripheral membrane proteins throughout the cell to allow for rapid re-palmitoylation and trapping at the Golgi[13, 27]. Consistent with Fyn palmitoylation at the Golgi, cells treated with the APT inhibitor, palmostatin-M[28], showed an initial accumulation of Fyn-mCit on the Golgi, followed by its dilution due to entropic equilibration to all endomembranes[27, 29] (Fig. 1f).

These data show that Src and Fyn are maintained at the plasma membrane via vesicular transport from the RE and the Golgi,

respectively. The next question we addressed is how these SFKs get to these organelles.

**UNC119 proteins are solubilisation factors for SFKs.** To address whether UNC119 solubilises myristoylated SFKs, we measured the direct interaction between mCitrine-tagged

SFKs and UNC119 proteins C-terminally fused to mCherry (UNCA-mCh or UNCB-mCh), by Förster resonance energy transfer (FRET) measured via fluorescence lifetime imaging microscopy (FLIM; Fig. 2a, b and Supplementary Fig. 3a, b). From the fluorescence decay profiles we calculated images of average fluorescence lifetime ($\tau$) and fraction ($\alpha$) of interacting

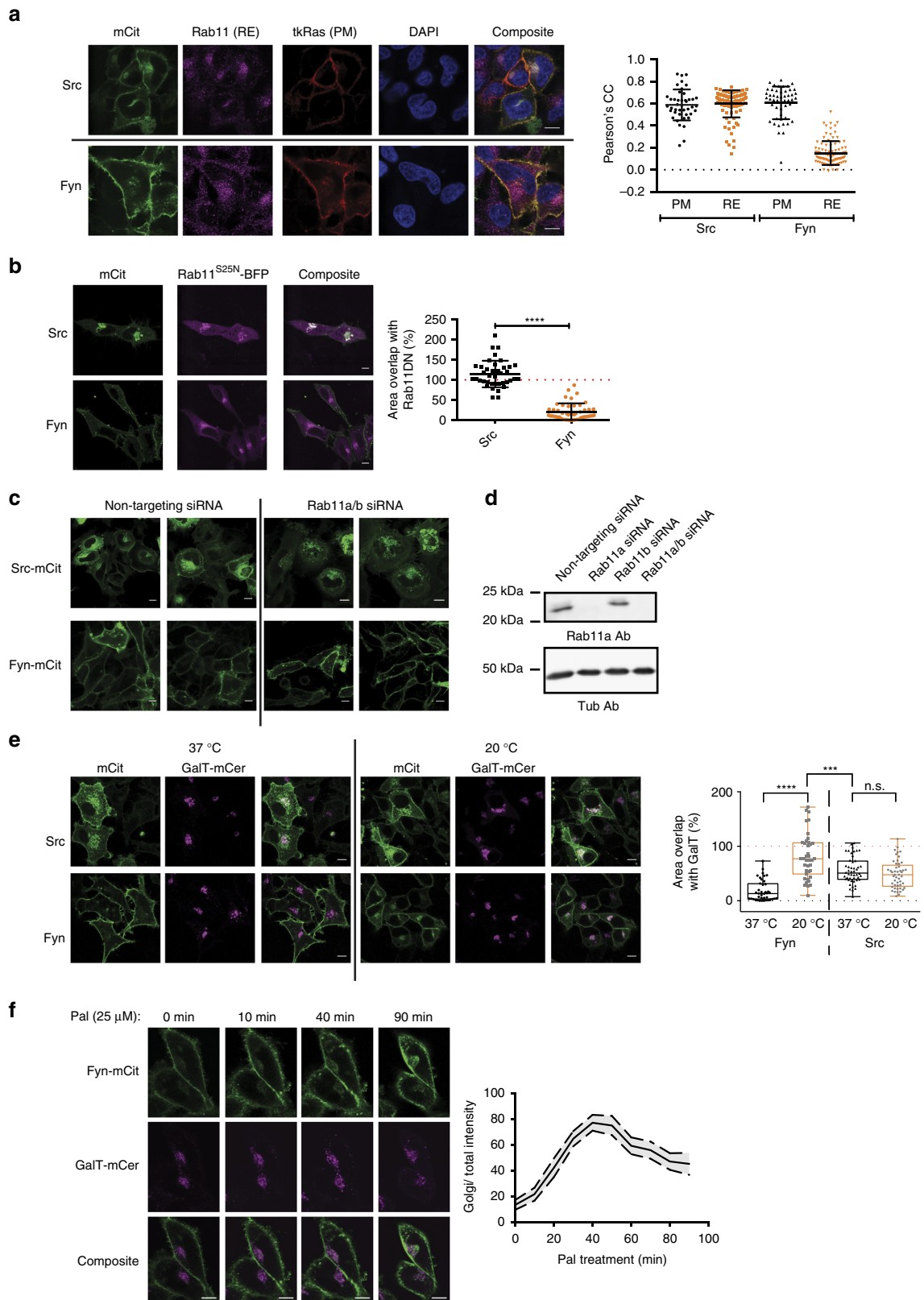

molecules that exhibited FRET[30, 31]. A decreased $\tau$ relative to the donor mCitrine alone ($\tau_{mCitrine} \approx 3$ ns) indicates the occurrence of FRET and interaction between the SFK and UNC119, the interacting fraction of which is represented by the parameter $\alpha$. Both UNCA-mCh and UNCB-mCh appeared homogenously cytosolic in HeLa cells, indicative of their soluble state (Fig. 2a, b, second column, and Supplementary Fig. 3a, b). The decrease in $\tau$ for cells that co-expressed Src-mCit and UNCA-mCh or UNCB-mCh clearly indicates the interaction between both proteins (Fig. 2a, first row, Fig. 2c and Supplementary Fig. 3a). From the radial profiles of $\alpha$ for Src-mCit interacting with UNCA-mCh it can be seen that it was maximal in the nucleus followed by the cytoplasm to decrease further at the periphery and the perinuclear region of the cell (Fig. 2a, g, 'Methods'). This shows that the low amount of Src in the nucleus (Fig. 2a, left column, and Fig. 2g) is maintained soluble by fully interacting with UNC119, whereas in the cytoplasm only a fraction of Src-mCit is interacting with UNC119. This is likely due to a membrane-interacting population of Src-mCit that cannot be solubilised by UNC119. This is supported by the observation that the fraction of interacting Src-mCit decreases in areas of the cell where the strongest association of Src-mCit with the negatively charged PM and RE occur. To validate that the strong association of Src with negatively charged membranes decreases the availability of Src that can be solubilised by UNC119, mutants of $Src^{nQ}$-mCit ($Src^{3Q}$, $Src^{4Q}$ and $Src^{6Q}$) with a reduced membrane dwell time were generated by altering the positively charged amino acids of Src's polybasic stretch to polar, uncharged Gln. These mutants distributed to all membranes in the cell due to their rapid exchange between membrane surfaces (Supplementary Fig. 3a). However, upon co-expression of both UNC119-mCh variants, these $Src^{nQ}$-mCits exhibited a more homogenous interacting fraction and solubilisation in the cytoplasm (Fig. 2a second row, Fig. 2d, h and Supplementary Fig. 3a). This solubilisation was not due to the lack of myristoylation, as apparent from metabolic labelling with YnMyr followed by click chemistry (Supplementary Fig. 3c). These experiments thereby show that the UNC119 proteins sequester Src that dissociates from membranes.

Fyn-mCit interacted with UNCA-mCh or UNCB-mCh primarily in the cytoplasm, with a clear reduced interacting fraction at the periphery of the cell (Fig. 2b, first row, Fig. 2e, f, i and Supplementary Fig. 3b). This suggests that palmitoylated Fyn that strongly interacts with membranes cannot be solubilised by UNC119. To test this, a non-palmitoylatable mutant of Fyn-mCit ($Fyn^{C3S-C6S}$) was expressed in HeLa cells. Similar to the $Src^{nQ}$-mCit mutants, these monolipidated $Fyn^{C3S-C6S}$-mCit also distributed to all membranes in the cell (Supplementary Fig. 3b, top right row). Upon co-expression of either UNC119-mCh variant, $Fyn^{C3S-C6S}$-mCit also exhibited a homogenous interacting

fraction leading to its solubilisation in the cytoplasm (Fig. 2b second row, Fig. 2e, f, j and Supplementary Fig. 3b). To resolve whether Fyn interaction with UNC119 is controlled by its palmitoylation, Fyn depalmitoylation was inhibited with palmostatin-M[28]. Palmostatin-M treatment gave rise to an equilibrated endomembrane distribution of fully palmitoylated Fyn, analogous to that of N/HRas (Supplementary Fig. 4a)[29]. This redistribution of fully palmitoylated Fyn was accompanied by a complete loss in interaction with UNC119 and hence no solubilisation (Supplementary Fig. 4a). We therefore conclude that UNC119 cannot interact with palmitoylated Fyn. In contrast, palmostatin-M treatment only marginally affected the distribution of non-palmitoylatable $Fyn^{C3S-C6S}$, which maintained its soluble state upon co-expression with UNC119 (Supplementary Fig. 4b). These experiments thereby show that UNC119 solubilises depalmitoylated Fyn that readily dissociates from membranes.

The interaction between non-myristoylatable Src-mCit or Fyn-mCit mutants ($Src^{G2A}$-mCit, $Fyn^{G2A}$-mCit) was not observable in cells expressing UNC119-mCh, providing a strong indication that the interaction occurs via the myristoyl tail of the SFKs (Fig. 2a, b, bottom row, Fig. 2c–f, Supplementary Fig. 3c and Supplementary Fig. 5a, b). To further substantiate this, we mutated a conserved tyrosine residue to alanine in UNC119-mCh (Y194A for UNC119A or Y205A for UNC119B) that reduces the affinity for myristoylated peptides by 140-fold[20] (Supplementary Fig. 5c)[32]. Neither SFK-mCit nor their mutants lacking their secondary motifs interacted with either $UNC119A^{Y194A}$ or $UNC119B^{Y205A}$ mutants in HeLa cells (Supplementary Fig. 5d, e). Finally, the interaction with UNC119 is not dependent on the kinase activity of Src, as a kinase dead mutant of $Src^{K298M}$ maintained the interaction with both UNC119 proteins (Supplementary Fig. 5f). This indicates that this interaction is a constitutive feature of the system that is uncoupled from Src activity.

To address the function of UNC119-mediated solubilisation of Src and Fyn, we investigated the localisation of both proteins upon knockdown of the UNC119 proteins. If UNC119 has a similar role for SFKs as PDEδ does for Ras in maintaining PM localisation, loss of its function should redistribute Src and Fyn to endomembranes. We therefore determined the co-localisation of Src and Fyn with the ER markers—Calreticulin-mCerulean (CalR-mCer) or Calnexin—after siRNA-mediated KD of UNC119, as the ER is the most extensive endomembrane system. KD of UNC119A/B in the non-tumorigenic breast epithelial cell line MCF10a as well as HeLa cells (Fig. 3c, h) resulted in redistribution of Src-mCit and Fyn-mCit to the ER (Fig. 3a, b and Fig. 3d–g). Similarly, UNC119 KD resulted in endogenous SFK relocalisation to endomembranes in MCF10a cells (Fig. 3i, j and Supplementary Fig. 6a). In order to address

**Fig. 1** Steady-state PM localisation of Src and Fyn requires vesicular traffic from the RE and Golgi. **a** Confocal images showing the steady-state localisation of mCitrine fused Src and Fyn in HeLa cells co-expressing the PM marker mCh-tK-Ras, and stained with antibodies against the RE compartment marker, Rab11a (see also Supplementary Fig. 2a). Dot plot depicts Pearson's correlation coefficient for Src and Fyn with either the PM marker or the RE marker ($n > 60$ cells per condition from two independent experiments, data are mean ± SD). **b** HeLa cells co-expressing the dominant-negative mutant form of Rab11 ($Rab11^{S25N}$-BFP) and either Src-mCit or Fyn-mCit (see also Supplementary Fig. 2b). Dot plot depicts the percentage of the area of Src or Fyn pixels staining internal membrane structures that overlapped with the $Rab11^{S25N}$-BFP pixels ($n > 40$ cells per condition, data are mean ± SD, ****$P < 0.0001$, Student's $t$-test). **c** Confocal micrographs of HeLa cells expressing Src-mCit and Fyn-mCit, transfected with Rab11a/b targeting siRNA or non-targeting siRNA control nucleotides. **d** Representative western blot showing Rab11a levels of transfected cells and the Tubulin-loading control ($n = 3$ independent experiments). **e** HeLa cells co-expressing the Golgi marker (GalT-mCer) and Src-mCit or Fyn-mCit, either cultured at 37 °C (left panels) or 20 °C (right panels) for 24 h, at which point cells were fixed. Graph depicts the percentage of the area of Src or Fyn pixels staining internal membrane structures that overlapped with the GalT-mCer pixels ($n > 40$ cells for each condition from two independent experiments, data are depicted as a box and whiskers plot, showing the median and the full range of the measurements. ***$P < 0.001$; ****$P < 0.0001$; n.s., not significant, as determined by Student's $t$-test. **f** Time series of HeLa cells co-expressing the Golgi marker (GalT-mCer) and Fyn-mCit, treated with the 25 μM of the APT inhibitor palmostatin-M (Pal). The plot to the right depict the integrated intensity of Fyn-mCit at the Golgi over the integrated intensity over the whole of the cell ($n = 17$ cells from two independent experiments; data are mean ± SEM). Scale bars, 10 μm

how solubilisation of SFKs by UNC119 maintains their PM localisation, we investigated the Arl-GTP-mediated release of SFKs from UNC119.

**Perinuclear release of UNC119 cargo is Arl2/3-dependent.** Structural and biophysical data indicate that UNC119 interacts with the small GTPases Arl2 and Arl3 in a GTP-dependent manner. This results in the allosteric opening of the UNC119 hydrophobic pocket[20] and the release of myristoylated Src peptides[33]. To first establish where Arl proteins reside in the cell, IF was performed in HeLa cells with Arl2-specific antibodies or Arl3-specific antibodies. This showed significant staining on the

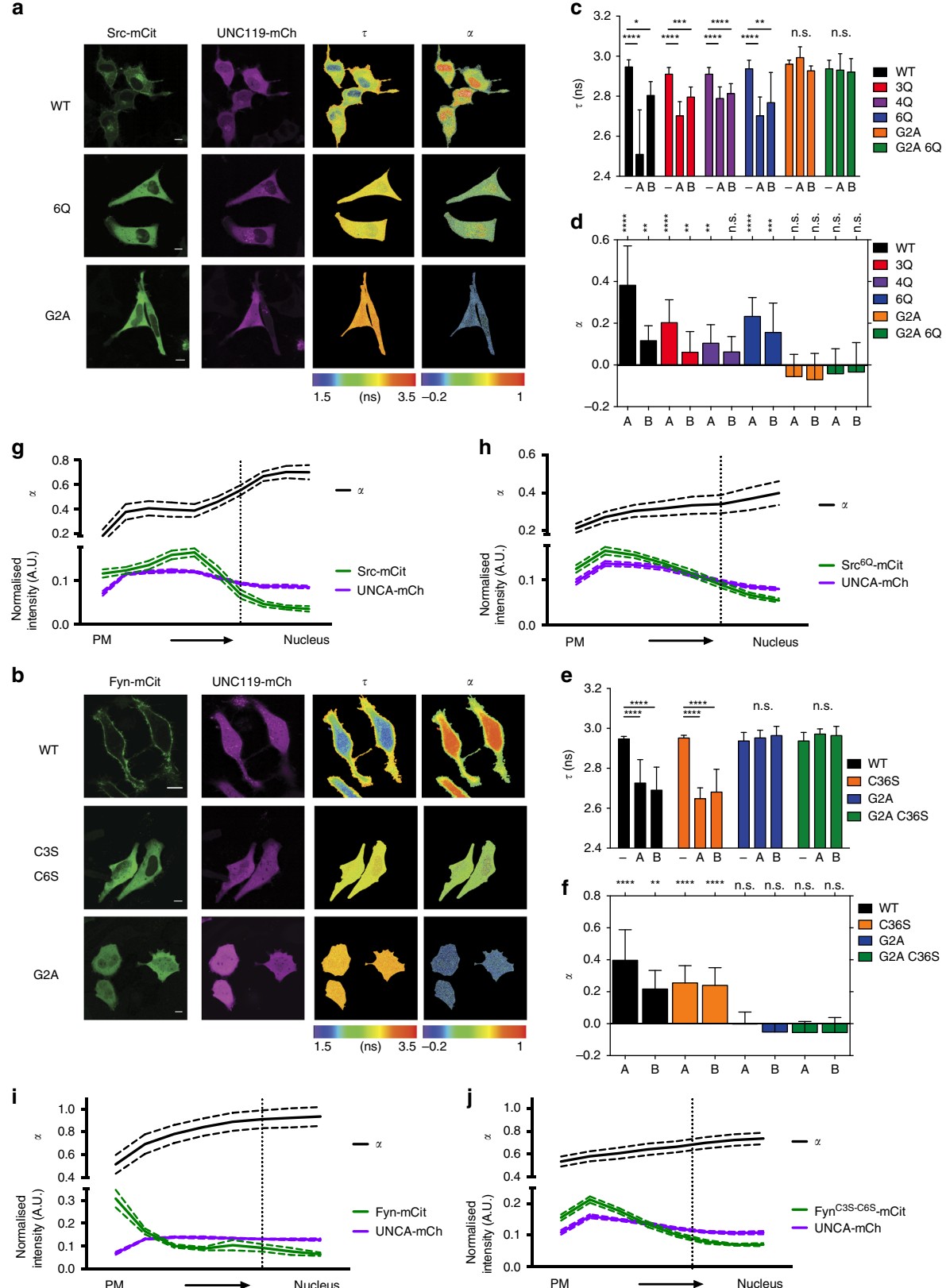

RE and proximal membranes, as well as in the nucleus and to a lesser extent in the cytoplasm (Fig. 4a and Supplementary Fig. 6b). This indicates that Arl proteins' partition between the cytosol and membranes, which is consistent with biophysical data that show that Arl proteins interact with membranes via their N-terminal amphipathic helices. For Arl3, the amphipathic helix is conformationally positioned for binding in the GTP state[34], which indicates that localised Arl3-GEF activity on acceptor membranes regulates this interaction. IF with an UNC119-specific antibody showed nuclear and cytoplasmic staining, with no obvious enrichment on membranes, consistent with the soluble state of the protein (Fig. 4a and Supplementary Fig. 6b).

To investigate whether and where this Arl-mediated release mechanism takes place in cells, we measured the interaction between endogenous UNC119 and Arl2/3 by in situ proximity ligation assay (PLA). The PLA reaction generates discrete fluorescent puncta in areas of the cell where protein interactions occur[35] (Fig. 4b). To obtain information on the radial distribution of this interaction, we computed the distance for each PLA punctum to the centre of the nucleus in many cells (see 'Methods', Fig. 4c). The puncta distributions for both Arl2/UNC119 and Arl3/UNC119 peaked at the position of the nuclear membrane and the adjacent cytoplasmic area, to rapidly decay towards the cell periphery. However, the shape of these puncta distributions is biased by the cell shape and size. To correct for this bias in the puncta distributions, a pixel–distance distribution to the centre of the nucleus that reflects the cell shapes was subtracted (Fig. 4d, 'Methods'). The positive peak around the average position of the nuclear membrane and the negative broad peak in the cytoplasmic area in these distance distributions shows that the interaction between UNC119 and Arl2 or Arl3 occurs predominantly on perinuclear membranes of the cell, indicating that SFK release from UNC119 occurs in this area.

We also examined the SFK/UNC119 interaction distribution as obtained from PLA. In contrast to the Arl/UNC119 puncta distributions, the SFK/UNC119 distribution peaked within the nucleus of the cells and exhibited a shallow, more even distribution throughout the rest of the cell (Fig. 4b–d, third row). This is consistent with this interaction occurring in the cytosol, as observed in our FRET-FLIM measurements (Fig. 2a, b).

To examine whether Arl-mediated release of Src and Fyn on perinuclear membranes sustains PM enrichment, the effect of siRNA Arl2/3 KD on Src-mCit localisation or Fyn-mCit localisation was investigated in HeLa cells (Fig. 5). Efficient KD of both Arl2 and Arl3 (Supplementary Fig. 6c) resulted in an increased Src-mCit-soluble fraction as apparent from the enhanced nuclear fluorescence (Fig. 5a). This was apparent in Arl2/3 KD cells that co-expressed UNC119, demonstrating impaired release of Src from UNC119 (Fig. 5a). The concurrent decrease in PM enrichment of Src-mCit (Fig. 5c) shows that Arl2/

3-mediated release of Src from UNC119 is necessary for efficient enrichment of Src on the PM.

In contrast to Src, Arl2/3 KD did not affect the spatial distribution of Fyn (Fig. 5b, c). This shows that Fyn localisation is dependent on the solubilising activity of UNC119 (Fig. 3), but not on its release from UNC119 by Arl-GTP. As only depalmitoylated Fyn interacts with UNC119 (Supplementary Fig. 4), it is likely that re-palmitoylation on the Golgi (Fig. 1f) is the dominant kinetic factor that concentrates Fyn on the Golgi after spontaneous, or Arl-mediated dissociation from UNC119.

**Released Src is trapped on REs to be transported to the PM.** We next addressed how the combined kinetics of Arl-mediated release of Src from UNC119 and its trapping on the RE, followed by vesicular transport, results in its enrichment on the PM. From fluorescence loss after photoactivation (FLAP) of Src C-terminally fused to photoactivatable GFP (Src-paGFP) on the RE and the concomitant gain of fluorescence at the PM, it is apparent that Src traffics from the RE to the PM (Fig. 6a). We quantified the kinetics of Src trafficking and diffusion by segmenting the cell in three regions (PM, RE and the communicating cytoplasm in between) normalised to the respective fraction per segment of Src-mCh fluorescence expressed in the same HeLa cell. This yields a measure of how far away Src-paGFP fluorescence is from the steady-state distribution of Src (signified by 1 in Fig. 6b, 'Methods'). In contrast to the relaxation of fluorescence at the RE and the cytoplasm after photoactivation, the fluorescence at the PM increases at a slow rate that is dominated by vesicular transport arriving from the RE and countered by endocytic processes from the PM. Fitting a mono-exponential recovery yields a rate constant in the range of $(7.9–8.9) \times 10^{-3} \, \text{s}^{-1}$ with 95% confidence. Assuming vesicular exit of Src-paGFP from the RE balances the rate of loss at the PM by endocytosis, this transit takes ~2 min. The subsequent fluorescence recovery after photobleaching (FRAP) of Src-mCh shows that the RE continuously sequesters Src from the cytosolic pool with an association rate constant $k_{on}$. Combining FLAP with FRAP, we extracted the exchange rates that give rise to the steady-state partitioning of Src on the RE. Modelling Src partitioning with two compartments (Src on the RE vs. the PM and the communicating cytoplasm), photophysical perturbation (FRAP or FLAP) of Src at the RE re-equilibrates in a single-exponential decay towards the steady-state partitioning determined by the ratio: $k_{on}/(k_{on} + k_{off})$ [36]. By adding a diffusible fraction, the data can be fit with $R^2_{adj} > 0.99$ (Fig. 6c, see 'Methods' and Supplementary Fig. 7 for fit without diffusion). From this, the recovery time of Src to the RE was determined to be ~2.5 min ($k_{on} = 5.8–7.4 \times 10^{-3} \, \text{s}^{-1}$, 95% confidence bounds (conf.)) and its residence time there to be ~1 min ($k_{off} = 1.5–1.97 \times 10^{-2} \, \text{s}^{-1}$, 95% confidence bounds (conf.)).

**Fig. 2** UNC119 solubilises SFKs in a myristoyl-dependent manner. **a**, **b** FRET-FLIM of the interaction between either mCitrine fused to Src **a** or Fyn **b** and UNCA-mCh in HeLa cells (see also Supplementary Figs. 3 and 4). For each sample, the fluorescence intensity of SFK-mCit (donor), the fluorescence intensity of UNCA-mCh (acceptor), the spatial distribution of the mean fluorescence lifetime ($\tau$) in nanoseconds and the molar fraction of interacting molecules ($\alpha$) are shown per the false-colour look-up tables. The *top rows* depict the steady-state localisation of wild type (WT) SFK-mCit when co-expressed with UNCA-mCh. The *second rows* show representative measurements with the polybasic stretch mutant Src[6Q]-mCit **a** or non-palmitoylatable mutant Fyn[C3S-C6S]-mCit (**b**), and the *bottom row* shows the non-myristoylatable mutant SFK[G2A]-mCit proteins when co-expressed with UNCA-mCh. *Bar graphs* on the *right* depict average $\tau$ (**c**, **e**) and $\alpha$-values (**d**, **f**) for different SFK proteins without (−), with UNCA-mCh (**a**), or UNCB-mCh co-expression (**b**; $n = 10–20$ cells/condition, data are mean ± SD from three independent experiments; *$P < 0.05$; **$P < 0.01$; ***$P < 0.001$; ****$P < 0.0001$ as determined by one-way analysis of variance (ANOVA) with Bonferroni *post-hoc* test). For $\alpha$ significance determination, samples were compared to the donor only control. Cells were sectioned into 10 segments of equal radius from the PM to the centre of the nucleus, and the mean intensity for each channel and $\alpha$ calculated for each segment (**g–j**), for the Src-mCit profile angular masking was performed to look at the profile change across the perinuclear RE compartment. *Line diagrams* depict the normalised intensity for each channel (SFK-mCit in *green* and UNCA-mCh in *magenta*) and the mean $\alpha$ (*black trace*) in each segment ± SEM; the *vertical dotted line* indicates the average localisation of the nuclear membrane ($n = 13$ cells for Src-mCit, Src[6Q]-mCit and Fyn[C3S-C6S]-mCit, and $n = 11$ for Fyn-mCit). *Scale bars*, 10 μm

This is faster than the fitted rate of vesicular transit from RE to PM. The obtained diffusion coefficient of 0.52–1.02 $\mu m^2 s^{-1}$ (95% conf.) is ~40 times too slow to represent free diffusion of UNC119-solubilised Src-molecules. Rather, part of the photo-activated/-bleached material diffuses on membranes that are not part of the RE out of the photophysically perturbed region (PPR). This process is aided by the solubilising UNC119 that facilitates

Src 'hopping' between membranes[14]. For this reason, ectopic expression of UNC119A-TagBFP increases the apparent diffusion to 1.137–1.382 $\mu m^2 s^{-1}$ (95% conf.) and halves the measured residence time in the PPR. That the recovery time of ~3 min to the RE ($k_{on} = 4.49$–$6.7 \times 10^{-3} s^{-1}$, 95% conf.) is only increased by 20% suggests that the Arl-GTP-mediated unloading of Src from the GSF operates close to saturating capacity. Therefore, ectopic

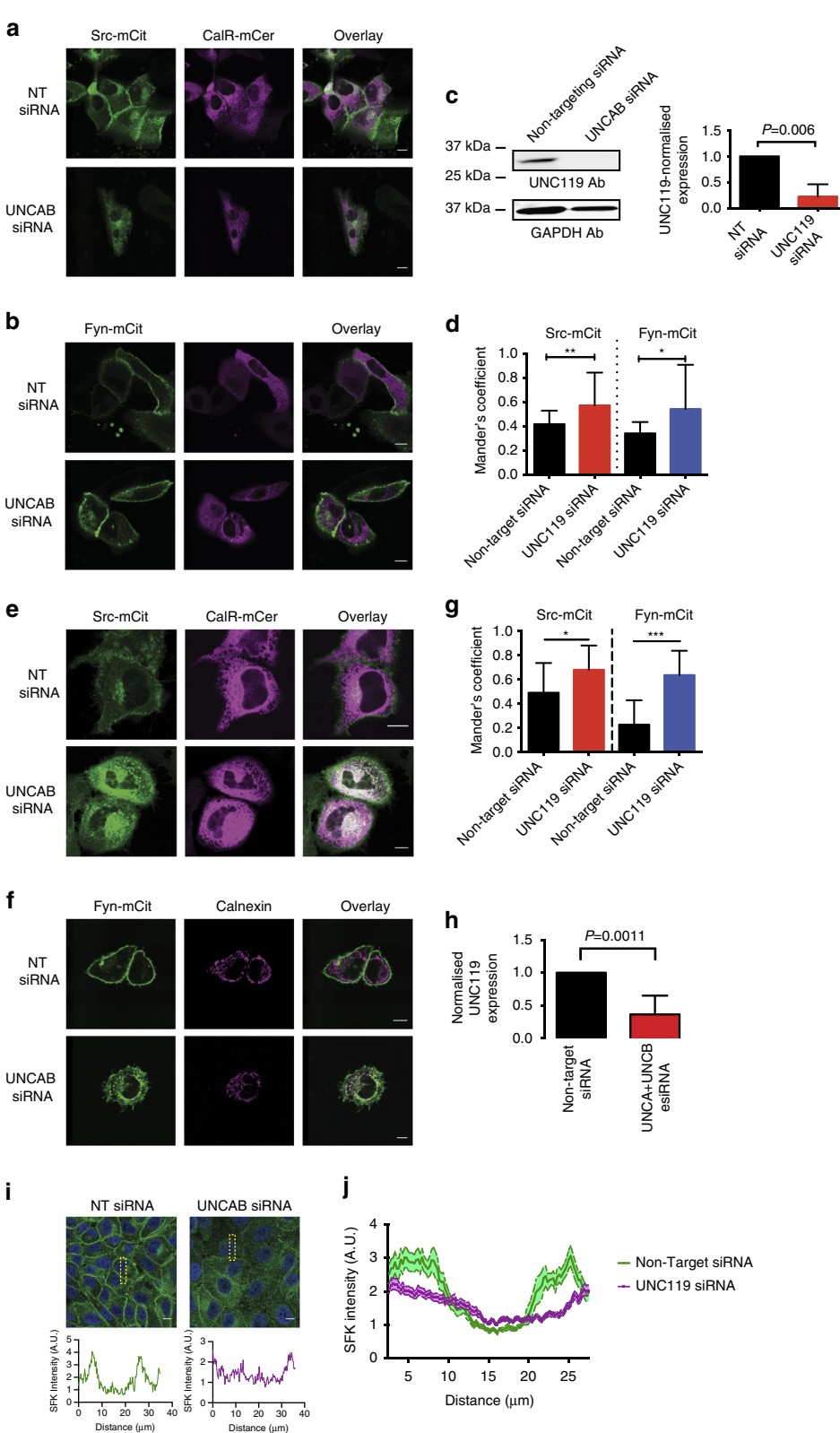

expression of UNC119 shifts the partitioning away from the RE towards endomembranes and a solubilised fraction. These data thereby show that the RE is a collector of solubilised Src released from UNC119 by rate-limiting Arl2/3 activity, to be redirected to the PM by vesicular transport.

**SFK-dependent colorectal cancer cell growth requires UNC119.** We investigated whether perturbation of SFK localisation by UNC119 KD affected the growth of SFK-dependent colorectal tumour cells. We chose to examine HT-29 colorectal carcinoma (CRC) cells, for which it has been shown that SFK activity is essential for their tumorigenicity[37–40]. We utilised a doxycycline-inducible Cas9 knockout system to generate UNC119 KD cells[41]. HT-29 cell lines stably expressing the nuclease Cas9 and doxycycline-inducible single guide RNAs (sgRNAs) targeting UNC119A or UNC119B or a control sgRNA sequence were produced by lentiviral transduction. This resulted in an average KD of 59% overall UNC119A/B upon UNC119A, and 84% upon UNC119B sgRNA induction (Fig. 7a). As the antibody recognises both UNC119A and UNC119B, we performed quantitative polymerase chain reaction (qPCR) measurements to quantify the KD level for each gene. UNC119A sgRNA induction resulted in a 73% KD of UNC119A and 38% of UNC119B transcript levels, while UNC119B sgRNA induction resulted in a specific 74% reduction of UNC119B levels (Supplementary Fig. 8a). The corresponding KD levels of overall UNC119A/B protein thus indicated that UNC119B was the predominant isoform in HT-29 cells, comprising at least 80% of total UNC119 expression.

UNC119 KD resulted in attenuated SFK activity in HT-29 cells as doxycycline induction of UNC119B or UNC119A sgRNAs resulted in a 57% and 28% reduction in Y419-SFK phosphorylation, respectively (Fig. 7a, b). UNC119 KD also decreased the Src-specific phosphorylation of Y925 on focal adhesion kinase (FAK)[42–44] by ~50% (Fig. 7a, b). Counterintuitively, overexpression of UNCA-mCh or UNCB-mCh also resulted in a decreased phosphorylation of Y419-SFK (Fig. 7c and Supplementary Fig. 8b). The dominant-negative effect of excess UNC119-mCherry can be explained by competitive binding to Arl2/3 of UNC119-mCherry not loaded with SFK cargo, resulting in a futile release cycle. This causes increased solubilisation and decreased PM association of SFKs (Fig. 5). In order to verify whether the decrease in SFK activity upon UNC119 KD is due to a loss of SFK PM enrichment, we examined the localisation of Src and mono-palmitoylated Yes, which are the predominant SFKs expressed in HT-29 cells, and are critical for their tumorigenicity[37–40]. Doxycycline induction of UNC119B KD resulted in the loss of PM enrichment of both Src-mCh and Yes-mCh, confirming that the PM localisation of all SFKs is dependent on UNC119 (Fig. 7d and Supplementary Fig. 8c). Indeed, Fyn-mKate2 PM enrichment was also reduced upon UNC119 KD (Fig. 7d and Supplementary

Fig. 8c), demonstrating that dually palmitoylated SFKs also require UNC119 activity to maintain their PM enrichment.

We next examined the effect of UNC119 KD on HT-29 cell proliferation by clonogenic assays (Fig. 7e). Overall proliferation as measured by covered surface area was only slightly reduced upon UNC119A sgRNA induction, whereas an almost complete lack of growth was observed upon UNC119B sgRNA induction (Fig. 7e, f). This is consistent with the lesser KD of total UNC119 levels upon UNC119A as compared to UNC119B sgRNA induction (Fig. 7b). To evaluate whether UNC119B KD affects cell death and/or proliferation, we analysed the number of clones and their size, respectively. UNC119B KD resulted in an 80% reduction in colony size and ~50% reduction in colony number (Fig. 7g, h). Despite that doxycycline-induced expression of the control sgRNA had a minor effect on cell growth, loss of UNC119-solubilising activity clearly affected both cell survival and proliferation.

## Discussion

We identified spatial cycles that dynamically maintain SFK localisation at the PM and thereby their signalling capacity. In these cycles, UNC119 proteins act as the GSFs by binding to and enhancing diffusion of myristoylated SFKs in the cytoplasm. The localised Arl2/3 activity unloads and concentrates the solubilised SFKs onto membranes on and proximal to the RE. There, electrostatic interaction with the negatively charged RE traps polybasic stretch containing Src, while palmitoyl acyl-transferase (PAT) activity concentrates palmitoylatable SFKs on the Golgi. The SFKs are then trafficked along the constitutive anterograde vesicular transport from these compartments, back to the PM (Fig. 8).

As the GSF PDEδ does for prenylated Ras proteins, UNC119 sequesters SFKs from the cytoplasm by binding their myristoyl tail within its hydrophobic cavity. This solubilisation competes with rebinding to membranes and allows slow lateral diffusion on endomembranes to be replaced by fast cytosolic diffusion. For Src, this enhanced diffusion increases the kinetics of electrostatic trapping at the RE. However, enrichment on these surfaces is also determined by the balance of available membrane surface area vs. the tighter binding due to electrostatic interaction. The latter proves insufficient to out-compete the vast surface area of endomembranes[14]. Instead, an energy-driven localised release of Src from UNC119 occurs to maintain out-of-equilibrium membrane enrichment. This is achieved by interacting with the small GTPases Arl2/3 in a GTP-dependent manner[18, 20] in the membrane-dense perinuclear region proximal to the RE. By this, Src is locally released from the GSF and associates to proximal membranes, thereby increasing electrostatic trapping at the RE. The out-of-equilibrium concentration of Src on the RE is transported to the PM by constitutive retrograde vesicular

**Fig. 3** UNC119-solubilising activity maintains SFK PM localisation. Confocal micrographs of MCF10a cells expressing Src-mCit (**a**) or Fyn-mCit (**b**) and the ER marker, CalR-mCer transfected with UNC119 targeting siRNA (*lower rows*) or non-targeting siRNA control nucleotides (*upper rows*). **c** Representative western blot showing UNC119 levels of transfected cells and the GAPDH-loading control. *Bar graph* depicts the quantification of UNC119 kD in which UNC119 levels were normalised to the tubulin-loading control and the non-targeting siRNA control ($n = 4$, data are mean $\pm$ SD; significance determined by Student's $t$-test). **d** Quantification of co-localisation with Mander's coefficient for SFK-mCit with the ER CalR-mCer marker in MCF10a cells ($n > 30$ cells per condition from two independent experiments; data are mean $\pm$ SD. *$P < 0.05$; ***$P < 0.001$; Student's $t$-test). *Confocal micrographs* of HeLa cells expressing Src-mCit **e** or Fyn-mCit **f** and the ER marker; CalR-mCer **e** or stained with anti-Calnexin antibodies **f**, transfected with UNC119 targeting siRNA (*lower rows*) or non-targeting siRNA control nucleotides (*upper rows*). **g** Quantification of co-localisation with Mander's coefficient for SFK-mCit with the ER markers for HeLa cells ($n > 15$ cells per condition from two independent experiments; data are mean $\pm$ SD. *$P < 0.05$; ***$P < 0.001$; Student's $t$-test). **h** Quantification of UNC119 KD in HeLa cells by western blot using UNC119 and tubulin antibodies. UNC119 levels were normalised to the tubulin-loading control and the non-targeting siRNA control ($n = 5$, data are mean $\pm$ SD; **$P < 0.001$; Student's $t$-test). **i** Confocal micrographs of MCF10a cells transfected with UNC119A and UNC119B targeting siRNA or non-targeting siRNA control nucleotides and then stained with an antibody against SFKs (*green*) and with DAPI (*blue*). Traces below show the vertical intensity profile plots of SFK for the *yellow dashed area*. **j** The mean SFK intensity profile plots for multiple cells ($n = 20$ cells per condition, mean intensity $\pm$ SEM). *Scale bars*, 10 μm

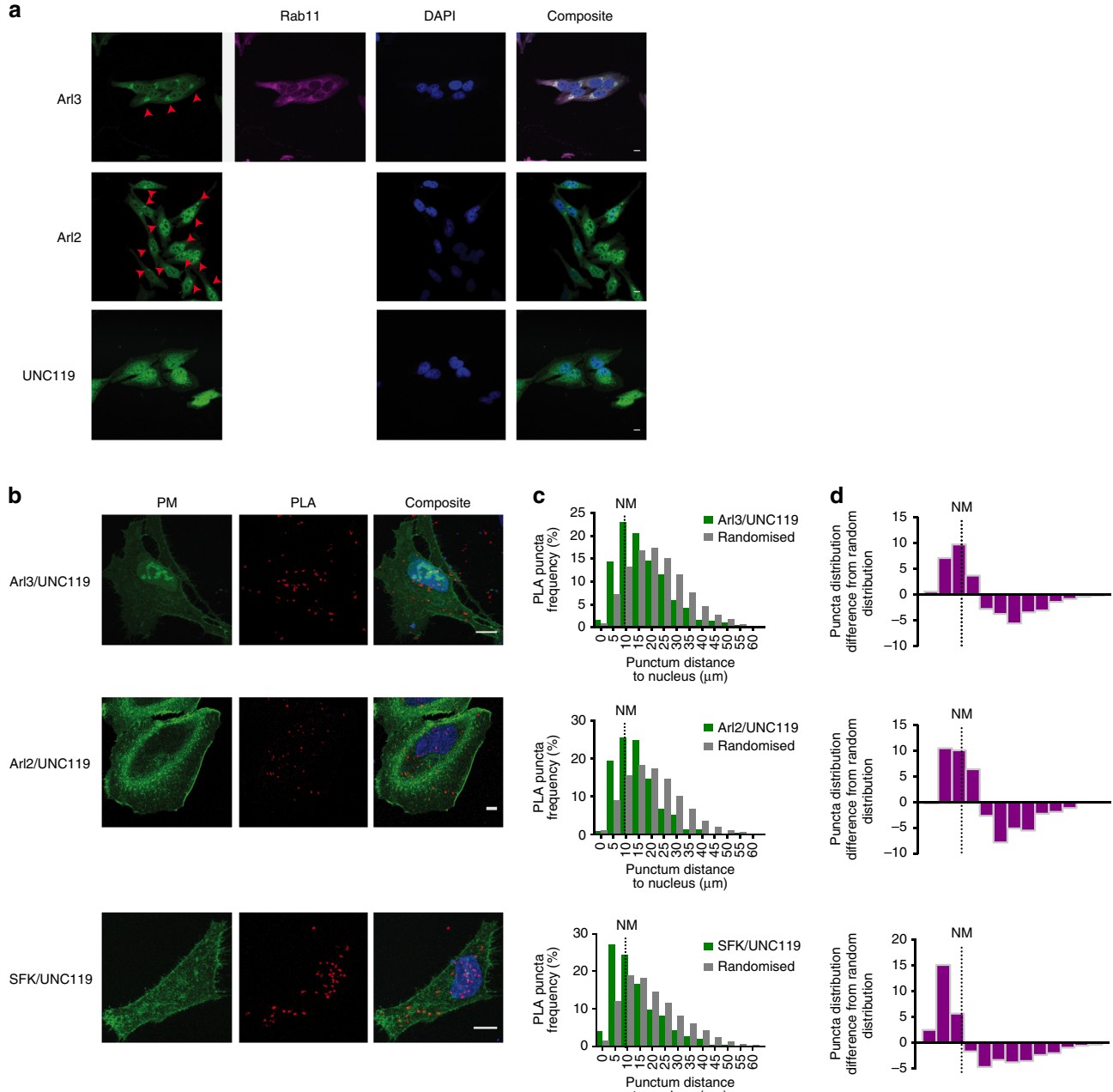

**Fig. 4** Arl2/3 interacts with UNC119 in the perinuclear area. **a** Confocal micrographs of HeLa cells stained with antibodies against Arl2, Arl3 or UNC119. Arl3-stained cells were co-stained with antibodies against Rab11. Nuclei were stained with DAPI (see also Supplementary Fig. 6b). *Red arrowheads* indicate the enrichment of Arl proteins on perinuclear membrane structures. **b** Maximum intensity projections of a confocal Z-stack of HeLa cells expressing the PM marker mCit-tk-Ras (*green*), in which in situ PLA (*red*) was performed using Arl2-, Arl3- or SFK-, and UNC119-specific antibodies to reveal the spatial pattern of endogenous interaction between these proteins. Nuclei were stained with DAPI. PLA puncta outside the cell periphery are from cells that were not transfected with mCit-tk-Ras. **c** *Histograms* depicting the mean distribution of distances of PLA puncta to the nuclear centre for the interaction pairs indicated (*green profiles*). This distribution is interleaved with a mean distance distribution of all pixels within cells to the nuclear centre (*gray profile*). **d** *Histograms* depicting the difference between the normalised PLA puncta distribution and the random distance distribution from **c** (n = 28 cells for UNC119/Arl2, 26 cells for UNC119/Arl3 and 47 for UNC119/SFK, data from at least two independent experiments per condition). *Scale bars*, 10 μm

transport from this recycling compartment. Our UNC119 overexpression studies show that UNC119 is not a direct activator of SFKs by releasing their intramolecular inhibitory interactions and/or stabilising the active conformation of the protein[7]. If this were the case, UNC119 expression should have led to an increased activity of SFKs instead of the observed decrease. Furthermore, direct UNC119-induced activation of SFKs is inconsistent with the fact that active SFKs are localised to the PM where they do not interact with UNC119. Instead, the highest

activity of SFKs would be expected in the cytoplasm where interaction with UNC119 occurs. The spatial cycle model does not contradict studies that have coupled the subcellular localisation of Src to its activity[45]. In this case, Src stimulation at the PM may drive its own transportation from the RE to the PM in a positive feedback involving RhoB activation[46, 47].

Fyn localisation is also dependent on UNC119-dependent diffusional exploration that enhances Golgi-localised PAT activity to kinetically trap depalmitoylated Fyn[13, 27]. From there, the

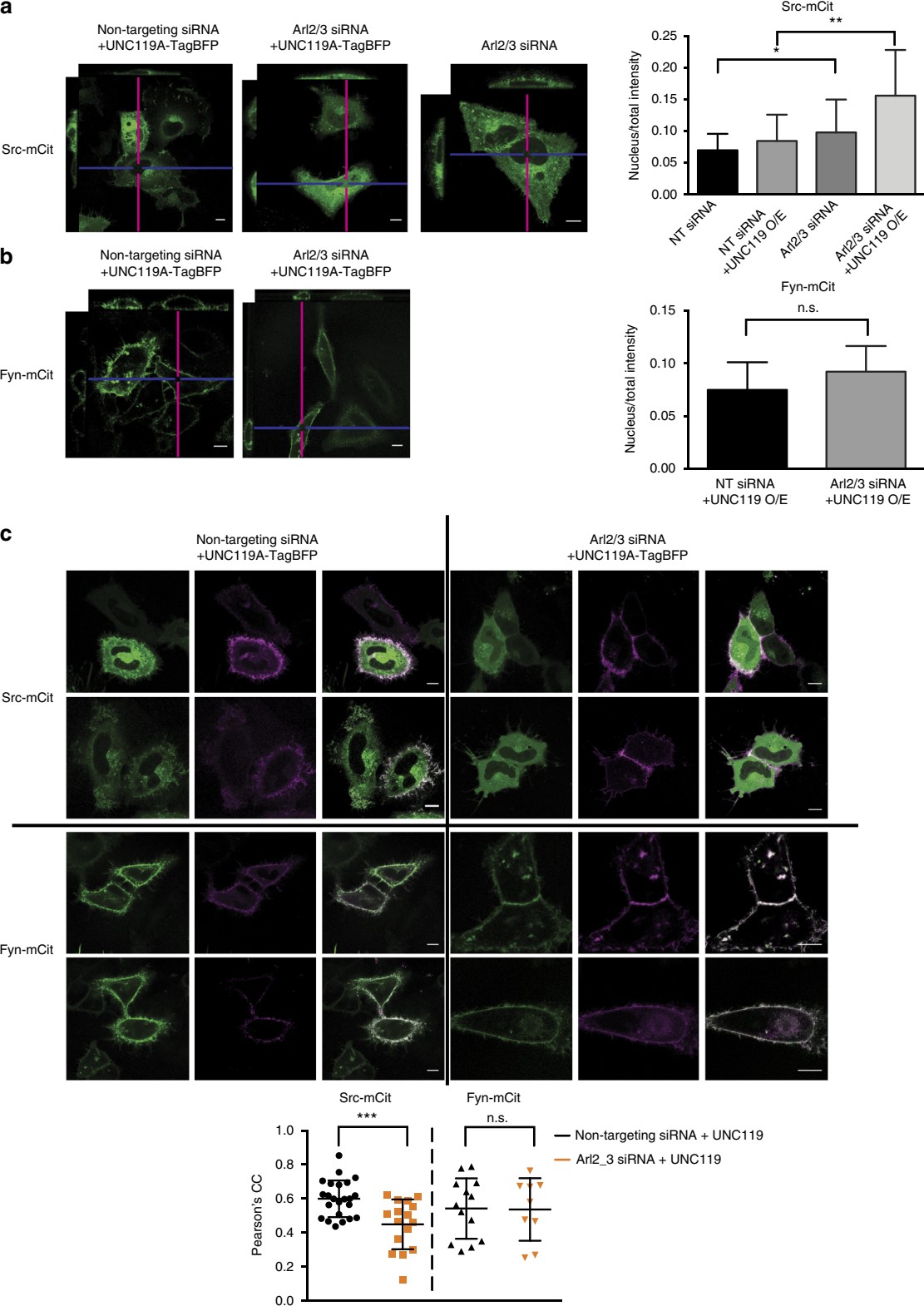

**Fig. 5** Arl2/3-mediated allosteric release from UNC119 maintains Src localisation. Confocal Z-stack projections of HeLa cells transfected with Arl2/3-targeting siRNA or non-targeting siRNA control nucleotides (see Supplementary Fig. 6c), ectopically expressing Src-mCit (**a**) or Fyn-mCit (**b**) with or without UNC119A-TagBFP expression. The intersection of the *magenta* and *blue lines* indicates the point in the stack that is depicted in the *xz* and *yz* planes. *Bar graphs* show the ratio of nucleus/total integrated fluorescence intensity for the SFK-mCit channel for each condition ($n = 10$–20 cells from three independent experiments; data denote mean $\pm$ SD; *$P < 0.05$; **$P < 0.01$; n.s., not significant; Student's $t$-test). **c** *Confocal micrographs* of HeLa cells transfected with Arl2/3-targeting siRNA or non-targeting siRNA control nucleotides, ectopically co-expressing Src-mCit or Fyn-mCit (*green*) and the PM marker mCh-tk-Ras (shown in *magenta*). *Dot plot* depicts the Pearson's correlation coefficient between SFK-mCit and mCh-tk-Ras ($n = 23$ or 16 cells for Src-mCit and $n = 13$ or nine cells for Fyn-mCit; data are mean $\pm$ SD from three independent experiments; ***$P < 0.001$; Student's $t$-test). *Scale bars*, 10 µm

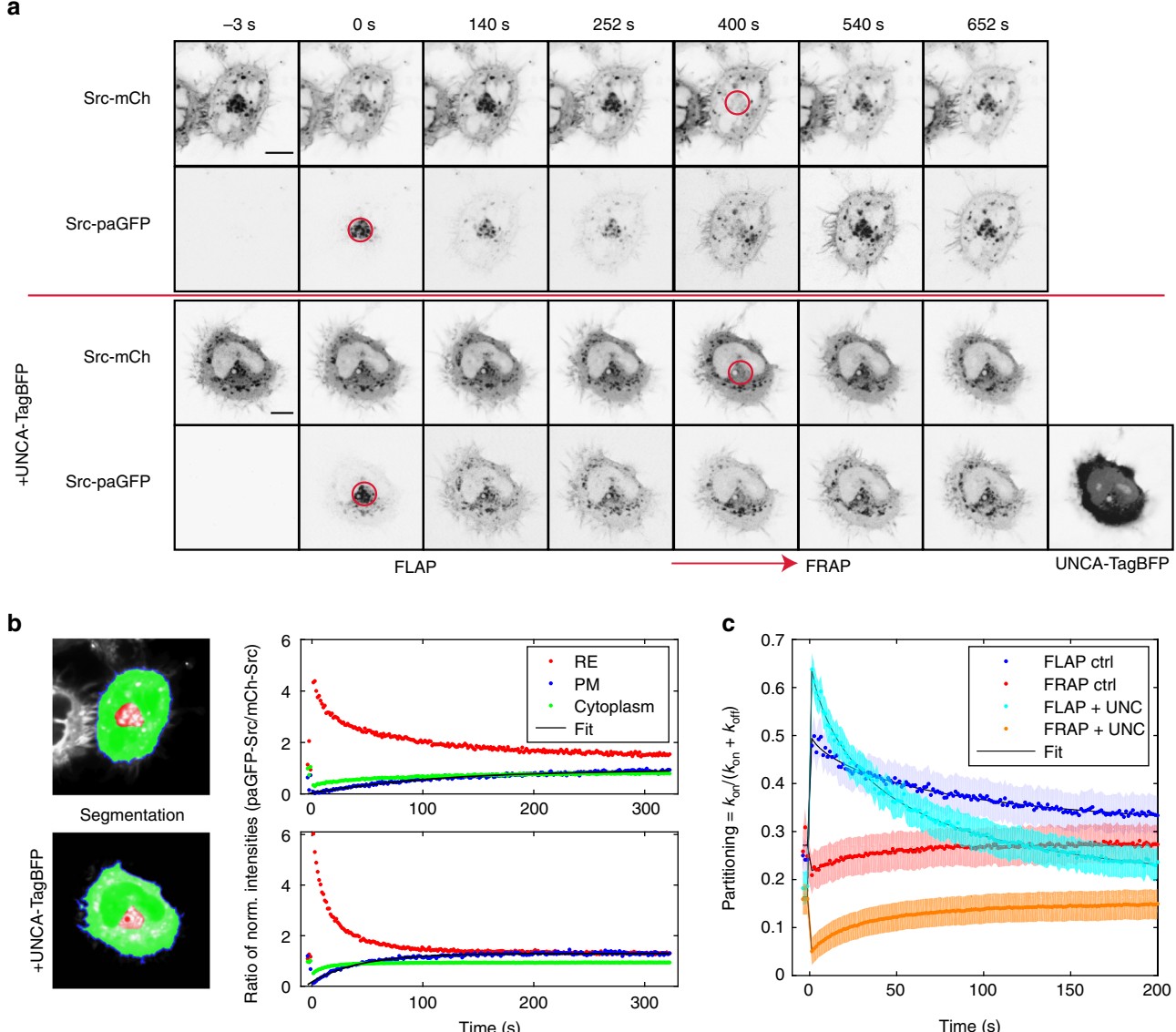

**Fig. 6** Src released from UNC119 by Arl2/3 is trapped at the RE to be transported to the PM. **a** Representative time-lapse sequences of fluorescence distribution of HeLa cells co-expressing Src-mCh (*first and third rows*) and SrcWT-paGFP (*second and fourth rows*) in the absence (*first and second rows*) or presence (*third and fourth rows*) of UNC119A-TagBFP ectopic expression. Src-paGFP was photoactivated at 0 s in the perinuclear region and subsequently Src-mCh was photobleached at 400 s in the same area (photoactivation/photobleach area: *red outline*). **b** Segmentation of cells in (**a**): RE (*red*), PM (*blue outline*) and communicating cytoplasm (*green*). The ratio of Src-paGFP fluorescence per segment (RE, *red*; PM, *blue*; cytoplasm, *green*) over total fluorescence in the cell and normalised to the respective ratio of Src-mCh fluorescence was plotted against time. The PM ratio was fitted to a mono-exponential recovery (*black line*). **c** Average traces of FLAP and FRAP curves fitted to the compartmental model described in Methods. Loss of Src-paGFP fluorescence at the perinuclear region normalised to whole-cell Src-paGFP fluorescence (*blue and cyan curves*) and corresponding gain of Src-mCh fluorescence after photobleaching at the same ROI, normalised to whole-cell Src-mCh fluorescence (*red and orange curves*). Control cells: mean ± SEM; $n = 5$ cells (*blue* and *red curves*), UNC119 ectopic expression: mean ± SEM; $n = 6$ cells (*cyan* and *orange curves*). *Scale bars*, 10 μm

secretory pathway reinstates PM enrichment. Arl activity has no influence on kinetic trapping at the Golgi because localised release of depalmitoylated Fyn from UNC119 on the RE cannot lead to electrostatic trapping because of the lack of a polybasic stretch on Fyn. Any Arl-mediated discharged Fyn on the RE will therefore quickly dissociate to get resolubilised by UNC119. Because Fyn will spontaneously dissociate from UNC119, it will continuously rebind and 'hop' between perinuclear membranes until it gets trapped and concentrated at the Golgi by palmitoylation.

The Src and Fyn spatial cycles are thus analogous to those of polybasic stretch containing KRas and palmitoylated H/NRas, respectively[15], which indicates that spatial cycles are a universal solution for countering entropic equilibration of lipidated peripheral membrane proteins to endomembranes.

A striking difference between the Ras and SFK spatial cycles lies in the specificity of the Arl-mediated release. Despite their homology, Arl3-GTP affinity for both GSFs is much higher than Arl2-GTP[34, 48]. These differences in affinity have been ascribed to the position of the N-terminal helix of Arl3-GTP, which is crucial for the displacement of high-affinity cargo in cilia[19, 20, 48]. However, the high-affinity displacement factor Arl3 should release high-affinity cargoes and low-affinity cargoes equally well. Therefore, Ras and SFKs should be equally displaced by either Arl from their respective GSF, as both have been shown to bind with

submicromolar affinity[49]. In line with this, we could show that the interaction of UNC119 occurs with Arl2 and Arl3 on perinuclear membranes proximal and of the RE. Accordingly, simultaneous Arl2/3 KD was required to disrupt the allosteric release of Src from UNC119. In contrast, Arl2 KD is sufficient to disrupt the allosteric release of KRas from PDEδ, indicating that Arl2 is the sole displacement factor for PDEδ[14]. These apparent differences in Arl specificity for the GSFs may reflect differences in the localisation of interacting binding partners[50] or local availability of their regulatory GEFs and GAPs[19, 51, 52]. Both Arl2 and Arl3 interact with membranes via their N-terminal amphipathic helices. However, for Arl3, this is dependent on GTP

loading of the protein[34], suggesting that its RE association is dependent on localised Arl3-GEF activity. Arl3 would thereby participate in a cytosol–membrane spatial cycle, with its membrane-bound form releasing myristoylated SFKs from UNC119.

SFKs have been implicated to have multiple roles in cancer progression[53, 54], including the promotion of drug resistance[55, 56] and maintaining cancer stem cell proliferation[57]. These illustrate the significance of inhibiting Src in combinatorial therapies[54, 55]. In CRC, Src activity increases with disease progression and is an indicator of poor clinical prognosis[58, 59]. HT-29 cells express high levels of active SFKs, specifically Src and mono-palmitoylated

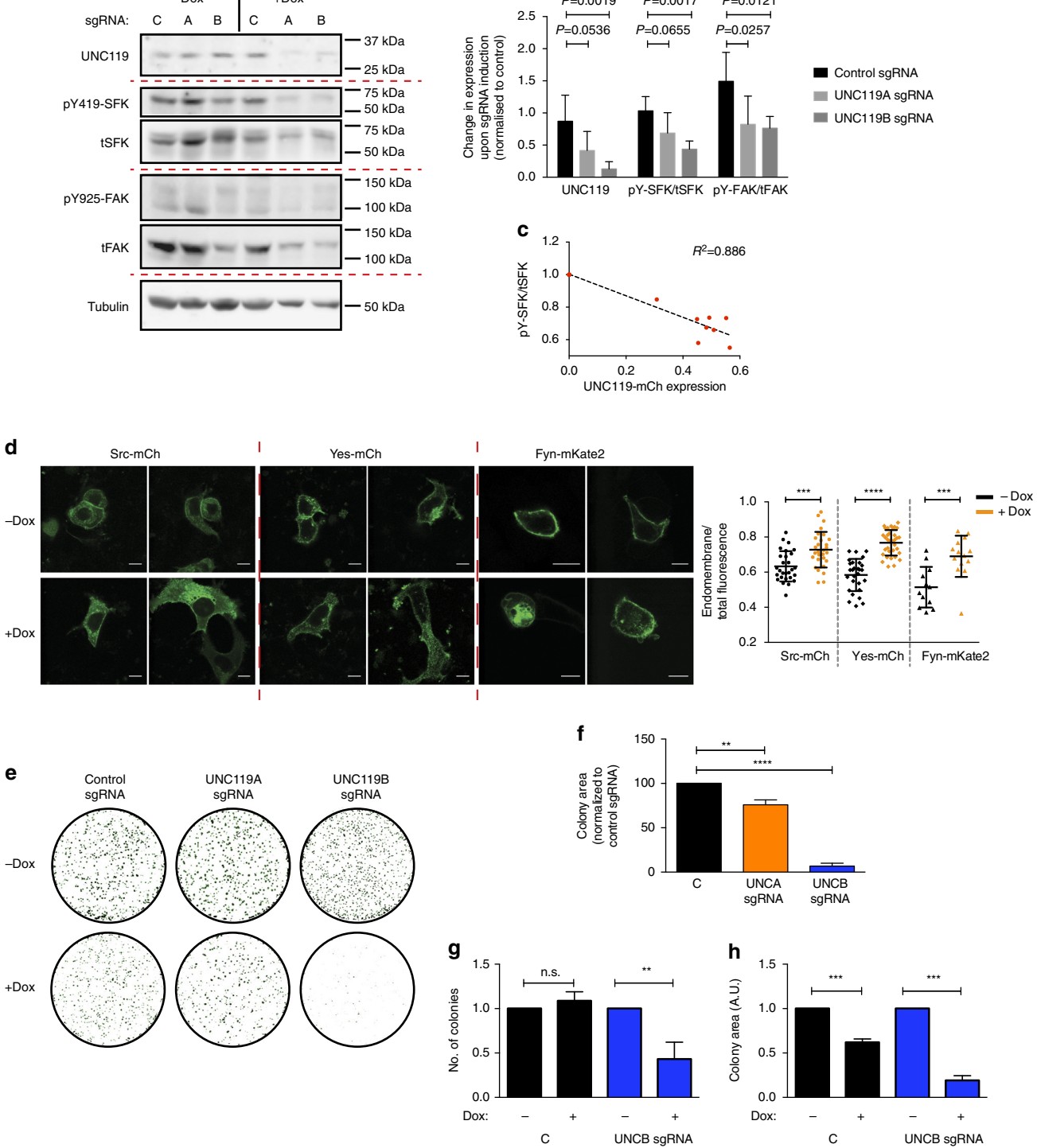

Yes[37–40]. These SFKs drive the tumorigenicity of these cells, although it is unclear whether a single SFK is important for this attribute[37, 38]. We have, however, demonstrated that UNC119 KD affects the localisation of both polybasic and palmitoylated SFKs, causing an overall reduction of their autocatalytic activation and signalling, resulting in strongly reduced proliferation of HT-29 cells. By interfering with the mechanism that maintains SFKs at the PM, we thus attenuated their activation and signalling output[9, 60, 61]. The pharmacological inhibition of the UNC119–SFK interaction, analogous to pharmacological inhibition of Ras interaction with PDEδ[31, 62], therefore could present a novel target to suppress oncogenic Src signalling.

## Methods

**Cell lines and transfections**. HeLa cells (ACC-57; Leibniz-Institut DSMZ-Deutsche Sammlung von Mikroorganismen und Zellkulturen GmbH, Braunschweig, Germany) were cultured in high glucose (4.5 g/l) DMEM supplemented with glutamine (PAN-Biotech GmbH, Aidenbach, Germany) and 10% fetal bovine serum (FBS, PAN-Biotech GmbH). MCF10a cells (CRL-10317; LGC Genomics GmbH, Berlin, Germany) were cultured in a DMEM/F12 culture medium (PAN-Biotech GmbH) supplemented with glutamine, 5% horse serum (PAN-Biotech GmbH), 20 ng/ml epidermal growth factor (Sigma-Aldrich Chemie GmbH, Munich, Germany), 0.5 mg/ml hydrocortisone (Sigma-Aldrich Chemie GmbH), 100 ng/ml cholera toxin (Sigma-Aldrich Chemie GmbH) and 10 μg/ml insulin (Sigma-Aldrich Chemie GmbH). HT29 cells (HTB-38; LGC Genomics GmbH) were cultured in Ham's F12 culture medium (PAN-Biotech GmbH) supplemented with glutamine and 10% FBS. All cells were grown at 37 °C in a humidified incubator under 5% $CO_2$ and regularly tested for mycoplasma. Cells were transfected at 80% confluence using Fugene 6 (Promega, Mannheim, Germany) according to the manufacturer's specifications. Rab11 siRNA transfections were carried out with Lipofectamine 2000 (Thermo Fisher Scientific, Dreieich, Germany) using 50 nM RNA. All other siRNA transfections were carried out with INTERFERin (Polyplus, Illkirch, France) using 50 nM of RNA.

**Plasmid construction and mutagenesis**. Restriction and modification enzymes were purchased from New England Biolabs (NEB, Frankfurt am Main, Germany). All PCR amplification reactions were performed with Q5 High-fidelity DNA polymerase according to the manufacturer's instructions (NEB). All PCR-derived sequences in the final constructs were verified by DNA sequencing.

mCitrine-N1/C1, mCherry-N1/C1, mTFP-N1, TagBFP-N1 and PAGFP-N1 were generated by insertion of AgeI/BsrGI PCR fragments of mCitrine, mCherry, mTFP (gifts from R Tsien), TagBFP (Evrogen) and PAGFP (gift from J Lippincott-Schwartz) complementary DNA (cDNA) into pEGFP-N1 or pEGFP-C1 (Clontech, Saint-Germain-en-Laye, France).

cDNA clones encoding full-length *UNC119A* (accession No: NM_005148), *UNC119B* (accession No: NM_001080533), *C-SRC* (accession No: NM_005417), *FYN* (accession No: NM_002037) and *YES1* (accession No: NM_005433) in the pCMV6 vector were obtained by BioCat (BioCat GmbH, Heidelberg, Germany). UNC119A, UNC119B, c-Src and Fyn constructs were made by PCR amplification from the cDNA clones using primers that introduced an EcoRI restriction site on the 5′ end and removed the stop codon followed by a AgeI or BamHI site on the 3′ end of the amplified cDNA fragment. Yes1 construct was made by PCR amplification from the cDNA clones using primers that introduced an NheI restriction site on the 5′ end and removed the stop codon followed by a KpnI site on the 3′ end of the amplified cDNA fragment. The PCR products were cut with EcoRI and AgeI or BamHI and subcloned into the mammalian expression vectors mCitrine-N1, mCherry-N1, mTFP-N1 and TagBFP-N1 (Clontech), which code for fusion proteins containing the different fluorophores mentioned at the C terminus. For Yes1, the PCR product was cut with NheI and KpnI and subcloned into the mCherry-N1 vector. All mutants were produced by Q5 site-directed mutagenesis (NEB) and were subsequently verified by sequencing.

MISSION esiRNA targeting human *UNC119A* and human *UNC119B* were from Sigma-Aldrich Chemie GmbH. ON-TARGETplus siRNA pools targeting human *ARL2* and *ARL3* were from Dharmacon (GE Healthcare Europe). Rab11a and Rab11b isoforms were simultaneously knocked down using siRNAs targeting the following target sequences 5′-AATGTCAGACAGACGCGAAAA-3′ and 5′-AAGCACCTGACCTATGAGAAC-3′, respectively[36].

**CRISPR/Cas9 and lentiviral transduction**. LentiCas9-Blast was a gift from Feng Zhang (Addgene plasmid #52962)[63], and FUCas9Cherry and the doxycycline-inducible sgRNA vector FgH1tUTG were a gift from Marco Herold (Addgene plasmid #70182)[41].

Twenty-four-bp oligonucleotides containing the sgRNA sequences were designed using the MIT CRISPR design website (http://crispr.mit.edu) and were synthesised (Sigma-Aldrich). The sgRNAs were cloned into the Bsmb-I site of the FgH1tUTG vector. sgRNA sequences are as follows: *UNC119A* exon 2: 5′-GGGGCTTCTTGATTTCAAAG-3′, *UNC119B* exon 1: 5′-GGGGCTGGTGGCTGGCAAGG-3′. Negative control: 5′-GACCGGAACGATCTCGCGTA-3′[64].

Lentiviruses were generated by co-transfecting 293T cells with 4 μg of Cas9 or sgRNA encoding plasmid and 4 μg of gag/pol and VSV-G env (pHITG) plasmids using Fugene6 (Promega). Growth media were exchanged the following day and lentivirus-containing supernatant was harvested 48 h later, concentrated using Lenti-X Concentrator (Clontech) and resuspended in the appropriate medium. HT29 cells were first transduced with the Cas9-containing virus, and were selected with 20 μg/ml Blasticidin S (Thermo Fisher Scientific) or FACS-sorted using mCherry expression. Selected single colonies were subsequently re-transduced with the inducible sgRNA vector containing UNC119 targeting sgRNAs. Cells were FACS-sorted and single colonies tested for KD of UNC119 by western blotting 72 h after induction with 2 μg/ml doxycycline.

**RNA extraction and qPCR**. Overall, $15 \times 10^4$ HT29 Cas9 cells were plated out per six wells, and cells were treated with or without 2 μg/ml doxycycline for 72 h. RNA extraction was performed using the Quick-RNA MiniPrep kit (Zymo Research Europe, Freiburg, Germany) according to the manufacturer's instructions.

The following primers were used for the subsequent qPCR: human *UNC119A* (sense, 5′-CAGGTTTAAGATTCGGGACATGG-3′; antisense, 5′-GGGCAACCGTTCTGAGACT-3′); and human *UNC119B* (sense, 5′-GCCGGGTCACCGAGAATTATT-3′; antisense, 5′-GAAACGCAAGGT TTGGCAATC-3′). KiCqStart SYBR Green Primers human *GAPDH* primers were from Sigma-Aldrich.

qPCR was performed using the GoTaq 1-step RT-qPCR system (Promega). Specifically, 15 ng input RNA was used in a 20 μl reaction in a 96-well thermal plate in triplicate. The plates were sealed and run in a iQ5 Real-Time PCR System cycler (Bio-Rad, Munich, Germany). Cycling conditions were as follows: 40 cycles of 95 °C for 10 s, 60 °C for 30 s and 72 °C for 30 s. Data were analysed using the ΔΔCt method for determination of relative gene expression by normalisation to an internal control gene (GAPDH), and fold expression change was determined compared to a control sgRNA sample.

**SDS-PAGE and immunoblotting**. Separation of proteins was performed by sodium dodecyl sulphate-polyacrylamide gel electrophoresis (SDS-PAGE), using 8, 12 or 15% Tris gels and Tris-glycine-SDS running buffer. Samples were prepared in

---

**Fig. 7** UNC119 KD in HT-29 CRC cells inhibits SFK activity and cell proliferation. **a** Representative western blot showing UNC119 expression level, total SFK (tSFK), total FAK (tFAK), tubulin (as loading control) and SFK phosphorylated on Y419 (pY419-SFK) and FAK phosphorylated on Y925 (pY925-FAK) in HT-29 cells expressing Cas9 and a doxycycline-inducible sgRNA vector, containing either a non-targeting guide RNA (**c**), or UNC119A (**a**) or UNC119B (**b**) targeting guide RNAs, induced with doxycycline for 72 h. **b** *Bar graph* depicting the mean change in expression upon sgRNA induction for pYSrc/tSrc, pYFAK/tFAK and UNC119, compared to the non-targeting sgRNA control (n = 5 independent experiments; data are mean ± SE. P values determined by one-way ANOVA with Bonferroni *post hoc* test). **c** HT-29 cells were transfected with UNCA-mCh or UNCB-mCh and probed for SFK phosphorylation by western blotting. Plot depicts pYSFK/tSFK normalised to non-transfected control against UNC119-mCherry expression normalised to GAPDH loading control from western blots (see also Supplementary Fig. 8b). **d** Fluorescence micrographs of HT-29 Cas9 cells expressing the doxycycline-inducible UNC119B sgRNA and treated with or without doxycycline for 72 h, transfected with Src-mCh, Yes-mCh or Fyn-mKate2 (see also Supplementary Fig. 8c). *Dot plot* depicts the endomembrane/total integrated fluorescence intensity for each SFK (n = 28 or 29 cells for Src-mCh, n = 28 or 34 cells for Yes-mCh and n = 12 or 14 cells for Fyn-mKate2; data are mean ± SD from two independent experiments, ***P < 0.001; ****P < 0.0001, as determined by Student's *t*-test). **e** Clonogenic assay of HT-29 Cas9 cells containing the indicated sgRNAs and induced with doxycycline for the duration of the assay. **f** *Bar graph* depicting the mean colony area coverage upon induction of sgRNA, normalised to the non-targeting sgRNA control well. **g** *Bar graph* depicting the mean colony number change upon induction of sgRNA, normalised to the pre-doxycycline control wells. **h** *Bar graph* depicting the mean individual colony size change upon induction of sgRNA, normalised to the pre-doxycycline control wells (n = 4 independent experiments; data are mean ± SE; **P < 0.01; ***P < 0.001; ****P < 0.0001; n.s., not significant as determined by one-way ANOVA with Bonferroni *post hoc* test). *Scale bars*, 10 μm

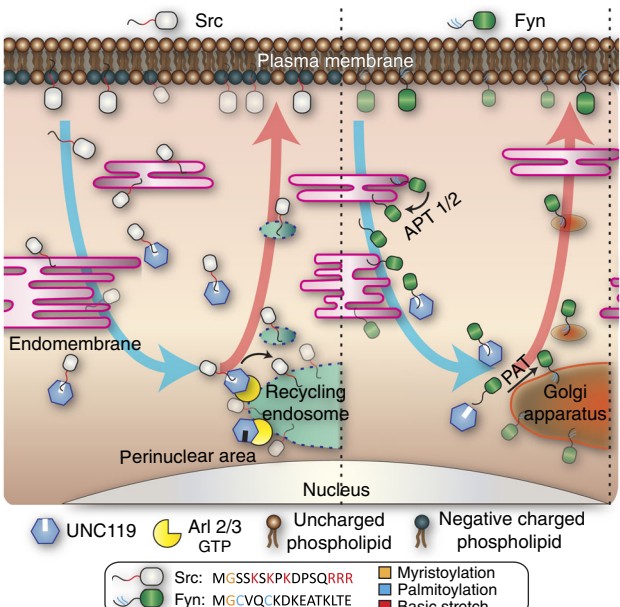

**Fig. 8** Spatial cycles of Src and Fyn counter entropic equilibration to endomembranes to maintain their PM localisation. SFKs are displaced by endocytic processes and spontaneous dissociation from the plasma membrane and equilibrate with the vast endomembrane system. Depalmitoylation by APT activity enhances dissociation of Fyn from membranes, whereas Src rapidly dissociates from the non-charged endomembrane surfaces. In the cytosol, the GDI-like solubilisation factor UNC119 binds to and enhances diffusion of the myristoylated SFKs. Arl2/3-GTP activity on and proximal to the recycling endosome discharges Src from UNC119 and concentrates it onto the RE where it gets trapped by electrostatic interaction with the negatively charged membranes (*scheme on the left*). In contrast, Arl2/3-GTP-mediated discharge of depalmitoylated Fyn from UNC119 on the recycling endosome cannot lead to electrostatic trapping because of the lack of a polybasic stretch in Fyn, resulting in rapid dissociation and resolubilisation by UNC119. Spontaneous dissociation of Fyn from UNC119 will result in membrane 'hopping' in this perinuclear area until kinetic trapping of Fyn at the Golgi by palmitoyl acyl-transferases catalysed palmitoylation (*scheme on the right*). The concentrated SFKs are then trafficked along the constitutive anterograde vesicular transport from these organelles, back to the PM. The *blue arrows* denote entropic equilibration to endomembranes; *red arrows* represent the countering, energy-driven relocalisation process

5× Laemmli sample loading buffer (10% SDS, 0.5 M Tris pH 6.8, 50% Glycerol, 0.1% bromo-phenol-blue) and 10% (v/v) 2-mercaptoethanol. The protein ladder used for comparison of molecular weight was Precision Plus Protein® Standards All Blue (Bio-Rad). Gels were run using the Mini-PROTEAN® Tetra Cell System and power supply unit (Bio-Rad).

Proteins were transferred from SDS-PAGE gels to a polyvinylidene difluoride (PVDF) membrane (Merck Chemicals, Darmstadt, Germany) using a semi-dry transfer unit (Bio-Rad). Membranes were blocked with Odyssey® TBS blocking solution (LI-COR Biosciences, Bad Homburg vor der Höhe, Germany) for 1 h at room temperature (RT) and washed with TBS-T (Tris-buffered saline, TBS, containing 0.05% Tween-20 (Sigma-Aldrich) and probed with appropriate antibodies. Antibodies and their sources were as follows: Mouse anti-UNC119 polyclonal antibody (H00009094-B01P, 1:200 dilution, Novus Biologicals, Abingdon, UK); Rabbit-anti-UNC119 monoclonal antibody (H00009094-K, 1:500 dilution, Abnova, Taiwan); Rabbit anti-UNC119 polyclonal antibody (GTX115185, 1:400 dilution, GeneTex, Biozol Diagnostica Vertrieb GmbH, Eching, Germany); Rabbit anti-Arl3 antibody (1:400, Proteintech, Manchester, UK); Rabbit anti-Arl2 antibody (1:2000, Abcam, Cambridge, UK); Rabbit anti-Src monoclonal antibody (36D10, 1:1000 dilution, Cell Signaling Technology Europe, Leiden, the Netherlands); Rabbit anti-phospho-SFK Y416 (D49G4, 1:1000 dilution, Cell Signaling); Rabbit anti-FAK monoclonal antibody (D2R2E, 1:1000 dilution, Cell Signaling); Rabbit anti-phospho-FAK Y925 (3284, 1:1000 dilution, Cell Signaling); Mouse anti-calnexin-monoclonal antibody (AF18, 1:100 dilution, Sigma-Aldrich);

Mouse anti-Tubulin monoclonal antibody (1:10,000, Sigma-Aldrich); Rabbit anti-GFP antibody (1:1000, Clontech); Rabbit anti-Rab11a (ab65200, 1:500 Abcam); and Mouse anti-GFP monoclonal antibody (1:1000, Clontech). Secondary antibodies used were as follows: donkey anti-mouse IgG-IR680; donkey anti-mouse IgG-IR800; donkey anti-rabbit IgG-IR680; and donkey anti-rabbit IgG-IR800 (1:5000, LI-COR); all secondary antibodies were infrared (IR)-labelled.

Visualisation was carried out by fluorescent detection on a LI-COR Odyssey CLx imaging system (LI-COR). Images were analysed with Fiji software[65]. Uncropped blots are shown in Supplementary Fig. 9.

**Metabolic labelling with alkyne-myristic acid**. HeLa cells were transfected with wild-type or mutant Src and Fyn constructs. Thirty-six hours post transfection, the medium was exchanged for feeding medium (DMEM, 3% FBS plus 50 μM alkyne-myristic acid analogue tetradec-13-ynoic acid (YnMyr) in DMSO, or the same volume of DMSO used as vehicle control). After 16 h, cells were rinsed twice with ice-cold phosphate-buffered saline (PBS) and then lysed with 100 μl lysis buffer (0.1% SDS, 1% Triton X-100, EDTA-free Complete protease inhibitor (Roche Diagnostics, Burgess Hill, UK) dissolved in PBS). Lysates were centrifuged at $16,000 \times g$ for 10 min to remove insoluble material. The supernatant was collected and used for further experiments.

Cell lysates (20 μg of total proteins) were reacted with CuAAC reaction cocktail containing azido-PEG-Biotin (AzB; 10 mM stock in DMSO) at 100 μM final concentration, CuSO4 (50 mM stock in water) at final concentration of 1 mM, TCEP (50 mM stock in water) at final concentration of 1 mM and TBTA (10 mM stock in DMSO) at final concentration of 100 μM. The reaction was vortexed for 1 h at RT, and EDTA (100 mM in water stock) was added to final concentration of 10 mM. Proteins were precipitated by addition of 4 vols of methanol, 1 vol of chloroform and 3 vols of water. The sample was centrifuged for 5 min at $16,000 \times g$ and the pellet was washed twice with RT methanol. The protein precipitates were resuspended in 2% SDS in PBS and further diluted to 1 mg/ml of total protein concentration and final SDS concentration of 0.2% with PBS. An aliquot of this sample was taken for SDS-PAGE analysis.

An amount of 200 μg of proteins from cell lysate was immunoprecipitated with 1 μg anti-GFP rabbit polyclonal antibody (Clontech). Following overnight incubation at 4 °C on a rotating wheel, 20 μl of protein-G magnetic Dynabeads (Thermo Fisher Scientific) were added to each sample and incubated for 1 h at 4 °C on a rotating wheel. The beads were washed (5×) with lysis buffer and resuspended in 20 μl of PBS, and freshly premixed CuAAC reaction reagents at appropriate concentrations (as described above) were added. After 1 h of vortex-mixing, the beads were washed (5×) with lysis buffer and 1× Laemmli reducing sample loading buffer and boiled at 95 °C for 10 min. Proteins were separated by SDS-PAGE and then transferred to PVDF membranes. Presence of myristoylated proteins was detected with IR dye-680 labelled Streptavidin (LI-COR) and precipitated proteins were identified by immunoblotting with anti-GFP mouse monoclonal antibody (Clontech) followed by IR800-labelled donkey anti-rabbit IgG (LI-COR) secondary antibody.

**Immunofluorescence**. Overall, $1.25 \times 10^4$ HeLa cells were plated on eight-well Labtech plates (Thermo Fisher Scientific) and transfected with Src-mCit and Fyn-mCit constructs. After 24 h, cells were fixed with 4% paraformaldehyde (PFA) in 1× PBS pH 7.5 for 10 min at RT, washed twice with 1× PBS containing 50 mM NH4Cl and twice with 1× PBS. Cells were then permeabilised in 0.1% Triton X-100/PBS at RT for 5 min. Rab11a was visualised with mouse monoclonal anti-Rab11 (1:100, Clone 47 610657, BD Biosciences) followed by AlexaFluor546-conjugated anti-mouse IgG (1:400, A10036, Thermo Fisher Scientific).

Confocal laser scanning microscopy was performed on a Leica TCS SP5 DMI6000 equipped with an HCX PL APO ×63 1.4 numerical aperture λ blue CS objective and an environment-control chamber maintained at 37 °C and 5% CO2. 4′,6-diamidino-2-phenylindole (DAPI)/BFP/mCerulean, mCitrine and mCherry fluorescence were excited using 405 nm Diode-UV laser, 514 and 590 nm WLL lines, respectively. Detection of fluorescence emission was restricted with an Acousto-Optical Beam Splitter as follows: DAPI/BFP, 425–450 nm; mCerulean, 470–500 nm; mCitrine, 524–551 nm; and mCherry, 610–650 nm. Scanning was performed in line-by-line sequential mode with ×4 line averaging. Confocality was controlled by limiting the pinhole size between 1.0 and 2 Airy units. Co-localisation analysis was performed using Fiji image analysis software. The fluorescence signals in the two channels were background-corrected by subtracting the mean intensity from a region with no cells. This mean intensity was set as a lower threshold. Regions of interest (ROIs) were then drawn around the transfected cells and then the Mander's and Pearson's co-localisation coefficients were determined from the resulting images in the two channels of each cell. Images from at least 60 cells per condition from two independent experiments were used in the analysis.

**Rab11$^{S25N}$-BFP and GalT-mCer co-localisation experiments**. HeLa cells were plated out in eight-well imaging plates and transfected as described above with equal amounts of either Src-mCit or Fyn-mCit and either the Rab11$^{S25N}$-BFP or the GalT-mCer construct.

Twenty-four hours post transfection (Rab11$^{S25N}$-BFP experiments) cells were imaged by confocal microscopy with the settings described above. For the GalT-mCer co-localisation experiments, 24 h post-transfection cell medium was changed

for complete DMEM medium containing 25 mM HEPES (Thermo Fisher Scientific) and cultured in a humidified incubator either set at 37 °C or set to 20 °C for a further 24 h. At this point cells were either fixed by 4% PFA and images acquired with a confocal microscope, or, the medium was exchanged for fresh DMEM medium without phenol red (Thermo Fisher Scientific) for live cell imaging at 37 or 20 °C and (5% $CO_2$) on a Leica SP5 confocal microscope.

Quantification of the Rab11$^{S25N}$-BFP co-localisation with Src-mCit and Fyn-mCit was performed in Fiji by defining two ROIs with intensity thresholding: the first for Rab11$^{S25N}$-BFP-labelled endomembrane structures of the cells, and the second for the SFK-mCit-stained structures within the cytoplasmic area of the cell. From these, the percentage of the SFK-mCit area that overlaps with the area covered by Rab11$^{S25N}$-BFP signal was computed. A similar analysis was performed for the GalT-mCer co-localisation experiments. For the analysis of the palmostatin M experiments, the integrated intensity of Fyn-mCit that co-localised with the GalT-mCer signal was normalised to total integrated intensity Fyn-mCit for the whole cell. Only cells that survived the full 90 min of treatment were considered for analysis.

**Proximity ligation assay**. In situ PLA was performed in HeLa cells to determine the interaction profiles of endogenously expressed UNC119 with Arl2/Arl3/SFK proteins. HeLa cells transiently transfected with the PM marker, mCit-tk-Ras, were fixed, permeabilised and blocked as described (see Immunofluorescence section), and different combinations of antibodies were incubated overnight at 4 °C. Antibodies used were as follows: Mouse anti-UNC119 polyclonal antibody (1:400 dilution); Rabbit anti-Arl2 antibody (1:1000 dilution); Rabbit anti-Arl3 antibody (1:1000 dilution); and Rabbit anti-Src monoclonal antibody (1:500 dilution). The next day Duolink in situ PLA (Sigma-Aldrich) was performed according to the manufacturers' instructions. Cells were then imaged by confocal microscopy, with the settings described above.

**FLIM**. FLIM experiments were carried out at 37 °C and 5% $CO_2$ on a Leica SP5 confocal microscope equipped with a Picoquant FLIM module (LSM Upgrade Kit/ SMD Module; Picoquant). mCitrine fluorescence was excited at 514 nm with a pulsed supercontinuum laser with a repetition rate of 40 MHz. Fluorescence signal was collected as described (see Immunofluorescence section above).

Photons from the mCitrine channel were detected using avalanche photodiodes and arrival times processed by a time-correlated single-photon counting module (PicoHarp 300; Picoquant).

Intensity thresholds were applied to segment the cells from the background fluorescence. Global analysis of FLIM-FRET data was implemented using Matlab R2014A (The Mathworks Inc., MA, USA) according to the process described in refs. [66, 67] to obtain images of the molar fraction (a) of interacting SFK-mCit with UNC119-mCh. Pixels with a total number of photons less than a pre-set threshold of 50 counts are excluded from the analysis.

**Segment analysis of FRET-FLIM images**. The segmenting analysis was performed with an in-house developed software developed in Anaconda Python programming language (Python Software Foundation, version 2.7, https://www.python.org/). Cells and nuclei were masked from the SFK-mCit channel images using Fiji. For each pixel within the cell, the distance to the closest PM and nuclear centre (NC) were calculated to derive a normalised distance $d = dPM/(dPM + dNC)$. All pixels were split into 10 intervals according to their normalised distances, giving rise to 10 cellular segments from the plasma membrane to the NC. For each channel the mean intensity was calculated for each segment, yielding a spatial profile for the individual cells. For angular masking, the centre of the nucleus was determined and the intensity-weighted 'centre of the cell' and a central axis was fitted to these two points. Fifty degree around the axis was used for the analysis of intensity distribution. For the intensity profiles, each value of the segment was normalised to the sum of intensity in all segments, and the resulting mean value with the SEM determined.5

**FLAP and FRAP studies**. FLAP and FRAP experiments were carried out at 37 °C and 5% $CO_2$ on a Leica SP5 confocal microscope with settings similar to those described above, with a ×63 oil immersion objective. Cells transfected with Src C-terminally tagged with a photoactivatable GFP (Src-paGFP) and Src C-terminally tagged with mCherry (Src-mCh) with or without co-expression of UNC119A-TagBFP were allowed to equilibrate in the environment–control chamber on the microscope. Images were acquired continuously every 1.29 s. After three 'pre'-images, fluorescence was perturbed over three frames for FLAP and five for FRAP by high-intensity 405-nm illumination in a mostly circular, perinuclear ROI that coincides with a substantial Src-mCh fluorescence in a steady state. The acquired images were background-corrected and the integrated ROI fluorescence was normalised to the integrated fluorescence intensity of the whole cell:

$$F(t) = \sum F(t)_{ROI} / \sum F(t)_{total}$$

This result provides the current partitioning of the SFK fluorescence and corrects for spurious photo-activation and photo-bleaching during image

acquisition. The distribution of SFK was considered as three compartments: (1) cytosolically diffusing, (2) inside the ROI and (3) outside ROI, with an association rate constant $k_{on}$ and a dissociation rate constant $k_{off}$ mediating exchange of material between (2) and (3), which is overlaid with a fast component in the initial phase after photo-physical perturbation. For the same cell, a sequence of FRAP was performed 4 min after a FLAP sequence was recorded. Both time traces were fitted to the generic function:

$$F(t) = \frac{k_{on}}{k_{on} + k_{off}} - \left( \frac{k_{on}}{k_{on} + k_{off}} - C_0 \right) e^{-(k_{on} + k_{off})t} + D_{cyt}(t)$$

where $C_0$ denotes the photo-physical perturbation (ideally $C_0 = 0$ for FRAP and $C_0 = 1$ for FLAP), $D(t)$ the cytosolic diffusion out of a circular ROI:[68]

$$D_{cyt}(t) = A_0 e^{-\frac{r^2}{2Dt}} \left( I_0 \left( \frac{r^2}{2Dt} \right) + I_1 \left( \frac{r^2}{2Dt} \right) \right)$$

with $I_0$ and $I_1$ as the standard modified Bessel functions, $D$ as the cytosolic diffusion coefficient, $r$ as the radius of the ROI and $A_0/C_0$ signifies the cytosolically diffusing fraction of photophysically perturbed SFK. The 'pre'-sequence partitioning of SFK should match the asymptotically reached steady-state partitioning $k_{on}/(k_{on} + k_{off})$ for both FRAP and FLAP sequences, where $k_{on}$, $k_{off}$ and $D$ were linked for the averaged FLAP and FRAP experiments in both cases (control and ectopic tagBFP-UNC119 expression). Results are shown in Supplementary Table 1.

Parameters of fluorescence increase at the PM after photoactivation in the perinuclear region fit to the equation:

$$F(t) = y_0 - (y_0 - A_0)e^{-kt}$$

Adjusted $R^2$: 0.976
Coefficients (with 95% confidence bounds):

- $A_0 = 1.87$ (1.84, 1.91)
- $k = 0.00838$ (0.00785, 0.00892)
- $y_0 = 0.954$ (0.934, 0.973).

**PLA distribution analysis**. PLA reactions occur between antibodies against the interacting proteins of interest resulting in discrete fluorescent puncta in areas of the cell where this protein interaction occurs. We quantified the distribution of the interacting PLA puncta relative to the nucleus within each cell in comparison to an estimated random distance distribution as a reference homogeneous distribution. The random distance distribution was estimated by calculating the distribution of the distances from each pixel within the cell mask to the NC. This procedure is analogous to adopting a Monte Carlo approach through redistribution of the PLA puncta randomly throughout the cell, i.e., randomly sampling several pixels within the cell (the same number as that of the PLA puncta) multiple times and averaging the resulting distributions. Consequently, the cell shape and size determine the random distance distribution and therefore have influence on the sensitivity of the identification of perinuclear localisation. Cells with a radius smaller than 25 μm and ones that had less than 10 PLA puncta were excluded from the analysis.

One-sided, one-sample Kolmogorov–Smirnov test was performed to test whether the PLA puncta data from each cell come as a sample from the random distance distribution or whether it peaks significantly closer towards the NC.

**Statistical analysis**. One-way analysis of variance with Bonferonni *post hoc* tests and Student's *t*-tests were performed in GraphPad Prism version 6.0e for Mac (GraphPad Software, La Jolla, CA, USA).

**Data availability**. The data that support the findings of this study are available from the corresponding author upon reasonable request.

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

## Acknowledgements

The project was partially funded by the European Research Council (ERC AdG 322637) to P.I.H.B. We kindly thank Christian Hedberg for providing palmostatin M, Christos Gatsogiannis for generating the UNC119 protein structure and Astrid Krämer for critical reading of the manuscript.

## Author contributions

P.I.H.B. conceived the project. A.D.K. cloned constructs, performed microscopy experiments, UNC119 RNAi and Cas9 experiments and PLA experiments. L.R. performed UNC119 and Arl RNAi experiments and cloned Cas9 plasmids. M.S. analysed the FLAP/FRAP data. A.S. analysed the PLA puncta distribution data. A.D.K. and P.I.H.B. designed the experiments and their analysis and wrote the manuscript with help of M.S.

## Additional information

**Competing interests:** The authors declare no competing financial interests.

