## [Peer Review File · Nature Communications]

Reviewers' comments:

Reviewer #1 (Remarks to the Author):

Konitsiotis et al.

The authors describe cycles of solubilization, trapping on perinuclear membrane compartments and vesicular transport that maintain the enrichment SFKs at the HeLa cell membrane which is essential for SFK signaling. Similar papers concerning Ras/Pdedelta/Arl2 were recently published (Schmick et al., 2014). The authors propose that UNC119 solubilizes myristoylated SFKs and enhances SFK diffusion to release SFKs on perinuclear membranes mediated by Arl2/3 GEF activity. Src, which cannot be palmitoylated, is trapped on the recycling endosome (RE) from where it traffics to the PM. Fyn is concentrated on the Golgi where it is palmitoylated by a protein S-acyltransferase, and traffics to the PM using the secretory pathway. UNC119 knock down disrupts SFK localization and signaling.

The authors use state-of-the-art technology (Crispr/Cas9, PLA, FLIM, FLAP and FRET) to quantitatively assay interactions and localization of UNC119 and SFKs in HeLa cells. The precise measurement of τ and α is quite impressive. However, determination of fluorescence intensities is at times difficult to follow, particularly looking at small insets as shown in fig. 1 or black/white representations throughout all figures. A concern is that all experiments were carried out only in HeLa cells. Are the experiments specific for tumor cells, and can they be repeated in non-tumor cells like hTert-RPE, IMCD3 or haemopoietic cells (Alam et al.)?

The model proposed by the authors calls for the RE (recycling endosome) to be a carrier trafficking SFKs back to the PM, and they identify REs using Rab11a antibodies (Figure 1). It is impossible to distinguish between RE and perinuclear membranes in DFigure 1, and it is not possible to see where Rab11a really is located. Rab11a is processed at the ER following prenylation and should be present there as well.

Abstract line 17: The authors state that "The solubilizing factor UNC119 sequesters mislocalized myristoylated SFKs from the cytoplasm, enhancing their diffusion to effectively release SFKs on perinuclear membranes by localized Arl2/3 activity." It is unclear why UNC119-solubilized SFKs are "mislocalized"? They should be soluble in the cytoplasm and therefore cannot be mislocalized. It is also unclear what the authors mean with "perinuclear membranes"? The rough endoplasmic reticulum is a perinuclear membrane. Is this what authors are referring to?

In the introduction (line 71), the authors state that the release of prenylated cargo on perinuclear membranes is mediated by a small GTPase Arl2 (or ARL3) in a GTP dependent manner. This would require that the ARL2- and ARL3-GEFs are located at the perinuclear membranes, as most ARL2/3 will be present in the inactive GDP form that cannot effectively unload myristoylated cargo from UNC119. The ARL3-GEF has recently been identified as ARL13b for which excellent antibodies are available. ARL13b, which is palmitoylated, is more likely present in the cell membrane.

The authors may wish to comment on biosynthesis of SFKs, which occurs most likely in the cytosol on free ribosomes. Cotranslational myristoylation is followed by binding to UNC119A/B to freely diffuse throughout the cell? If so, the location of their GEFs is very important. The GEFs will catalyze GDP/GTP exchange and enable unloading of cargo to the GEF containing membrane. Spontaneous dissociation (line 60) from the membrane should then not matter as SFK-binding to the membrane can be replenished from the pool of soluble SFK/UNC119.

Other Comments

Line 81. Ref 20 is incorrect as Kobayashi et al only showed interaction with ARL2 which is not based on a hydrophobic pocket.

Line 107. "Src-mCit was enriched at the PM and consistently (97% of the cells, n=38) at the Rab11a positive recycling endosome." This is difficult to see. SRC seems to be cytoplasmic in 1a, same for tk-Ras-mCherry. Color for mCit and mCherry may help to localize the proteins. Fyn-mCit localization to the PM is much clearer.

Line 129. " Src-mCit interacted with both UNC119A-mCh and UNC119B-mCh in the cytoplasm, but

not at the PM and the perinuclear region of the cell (Figure 2a-i,-iv,-v..". According to Fig. 2A, the strongest τ and the smallest α are at the nuclear region (or is the perinuclear membrane?), not in the cytoplasm.

Lines 151-162: Figure 3a and 3d are never mentioned in the text.

Line 165. "Structural and biophysical data indicates that UNC119 interacts with the small GTPases Arl2 and Arl3 in a GTP-dependent manner." A GEF is necessary to catalyzes GDP/GTP exchange. For ARL3, the GEF is membrane bound, and could well be in the PM. ARL3-GTP generated there could well be the trigger for unloading myristoylated cargo. The authors may want to localize ARL13b in HeLa cells, to determine where cargo is unloaded.

Line 182. "Thus, Arl2/3 activity is localised to the perinuclear region, which can lead to a concentration of myristoylated cargos on membranes in this area by localised release from UNC119." If the ARL3 GEF can be localized to the perinuclear region, this would be acceptable.

Line 221: maybe replace "is gained" by "increases"

Figure 1: The overlays are too small. Would it not be better to show b/w figures in color? PM localization of src is not easy to see. Fyn at Golgi is also not convincing.

Fig. 2A. The protein interaction between UNC119 and SFK was determined by FRET-FLIM. In Fig. 2A, why is the shortest average fluorescence life time (strongest interaction) at the nucleus (blue color in τ image) not mentioned in the text?

Fig. 3a describes the siRNA knockdown of UNC119ab compared with non-targeting siRNA. The cells look very different in I and ii, why? While Src-Cit PM localization is decreased, its concentration at the perinuclear region does not change. One would expect that without UNC119, Src would distribute to the whole endomembrane system, as Fyn-Cit does in Fig. 3b ii.

In Fig. 3a ii, the authors show Src and ER marker CalR co-localize. Where is Rab11? Is Rab11 colocalizing with SRC? If it does, UNC119 is not necessary for location at the RE and trafficking of SRC to the PM.

Fig. 4a, FRET-FLIM showed Arl2/3-UNC119 interaction throughout the cytoplasm (text lines 166-169). As ARL2/3 interact with UNC119 in a GTP-dependent manner, this would mean ARL-GTP is distributed all over the cytoplasm and thus release of Src from UNC119 could occur throughout the cytoplasm! This is in contradictory to Fig. 4b-c results showing Arl/UNC119 interaction concentrated in perinuclear region.

Fig. 4c. ii is not clear how the randomized puncta distribution (green bars) was generated. It's confusing why the distribution pattern is different between Arl/UNC119 (peak around 20um) and Src/UNC119 (peak around 15um). Because the randomized puncta in Src/UNC119 PLA quantification peak around the nuclear area (15um), these cells are not suitable for analysis, as the data could also interpreted as Src/UNC119 interaction occurs perfectly at perinuclear region (15um from center, similar to Arl/UNC119). The randomized puncta distribution pattern depends both on the relative place of nucleus within the cells (central vs peripheral) and cell size/shape. Ideally, the authors should use those cells with central nucleus for quantification so that the generated randomized puncta distribution pattern is not center around nucleus. Also, a Src/UNC119 PLA fluorescent image should be provided for Fig. 4c. Finally, Fig. 4c legend states "cumulative distribution of distance of PLA puncta....." (lines 862-863), but this is not cumulative distribution. If it's cumulative, the percentage will increase with distance and reach 100% at maximal distance.

Figure 5: IntDen? Total integrated fluorescence intensity? The figure is confusing as something appears to be below the panels in Fig. 5. What is the redline going through the pictures?

Fig.5b/ Fig.6 interpretation and discussion (lines 218-327) are confusing. As UNC119 is obviously important for Fyn PM localization (Fig.3b), there must be solubilized Fyn-UNC119 in the cells. The authors claimed that solubilized Fyn-UNC119 is directly trapped to Golgi by PAT enzyme but not released by Arl2/3 at perinuclear region. There was no evidence provided in this paper to support this claim. Also, it's unclear why the Fyn-UNC119 shows preference for Golgi PAT but not perinuclear Arl2/3, and whether Fyn need to be released from UNC119 before palmitoylation at Golgi (and if so, what the mechanism?).

Fig. 7, the authors claimed that released Src is trapped to RE and subsequently transported to PM by vesicular trafficking, but evidence is lacking. To exclude the possibility that released Src is also

trapped at other organelle (like ER), the author need use additional appropriate experiments (such as Rab11a RNAi, Transferrin labeling etc) to verify that RE is indeed the site of Src partitioning.

Reviewer #2 (Remarks to the Author):

The authors describe an original mechanism of protein tyrosine kinases Src and Fyn intracellular trafficking. They provide evidence for spatial cycles of solubilisation, trapping on perinuclear membrane compartments and vesicular transport that maintain the enrichment and activity of expressed fluorescence Src and Fyn molecules at the plasma membrane. They also identify the solubilizing factor UNC119, a previously identified Src interactor and activator, as an important component of this molecular process. Functional evidence for this mechanism is provided by a reduced Src activity upon UNC119 silencing in tumor cells, which correlates with a reduced cellular growth in vitro.

The proposed model is potentially interesting; however there are several concerns about the experimental design and on the role of this process in Src signaling that renders the conclusion less convincing.

Major points

1. Data incorporated in this report relies on Src (or Fyn) constructs fused to a fluorescent sequence at the C-terminus. These fusions are expected to stabilize the enzyme in an open and active form, unless a linker is inserted; therefore, the authors need to address whether these constructs alter Src catalytic activity and/or confirm the proposed model with a regulated Src fusion.
2. Most data incorporated in this report relies on overexpressed SFKs molecules, which might force the observed trafficking mechanism. Can the authors provide experimental evidence with endogenous SFKs?
3. It is not clear how this mechanism is regulated. UNC119 was previously described as a Src activator, which raises the question of the role of Src kinase activity in this trafficking process.
4. The authors should address how this trafficking function of UNC119 can fit with its previously reported Src regulatory/signaling function. How Src signaling induced by receptors cope with the proposed mechanism?
5. Functional data supporting an important role for this molecular process is not convincing. UNC119 may interact with several cargos in tumor cells in addition to SFKs; therefore the cellular effect observed upon UNC119 depletion is not sufficient to conclude to an important role of Src-UNC119 trafficking on Src oncogenicity. Besides, Yes seems to play a more important role on HT29 cell transforming properties than Src; however the authors did not address the role of UNC119 on Yes trafficking.

Minor points

1. Presented data supporting a role for UNC119 on Src trafficking impacting on Src activity is weak; therefore the title should be corrected accordingly, otherwise more experimental data supporting this notion should be incorporated in the ms.
2. The authors may include a paragraph on what is already known on SFKs trafficking. This would clarify why they became interested in Fyn and why they observed a different trafficking pathway between these kinases.
3. To my opinion, pictures shown in Figure 1 are too small to have a clear idea of where Src is distributed.
4. Does this UNC119-dependent trafficking route is used by Src oncogenic alleles?
5. It is not clear why the designed Cas9 strategy induces a partial KD of UNC119 proteins. On the same line, it is not clear why they overexpress Fyn, which is not linked to colorectal tumor malignancy.
6. The effect of UNC119 KD on Src activity can be explained many ways other than a defect on Src trafficking.
7. Anti-pTyr418 antibodies recognize all SFKs and this point should be corrected accordingly. On

the same line, WB incorporated in Fig 8 is puzzling as HT29 cells express aberrant activity of many SFKs while the blot shows one major band. Finally, I would suggest to confirm the effect on UNC119 on SFK signaling by measuring the phosphorylation level of another substrate.

8. Does UNC119 overexpression augment Src transforming properties?

9. The paper is well written and statistical analysis of the experimental data is sound; however some references on Src trafficking and the role of Src in colorectal cancer are missing. Rather, they authors quoted interesting papers on the role of Src in intestinal homeostasis and adenoma formation.

Reviewer #3 (Remarks to the Author):

Reviewer comments

The manuscript by Konitsiotis, et al. extends prior work from the Bastiaens group to quantify mechanisms that control the membrane localization of peripheral membrane proteins. Similar to prior work on Ras, the authors argue that two Src family kinases (SFKs, Src and Fyn) require solubilisation and trafficking mechanisms in order to be enriched at the plasma membrane (PM) in an out-of-equilibrium fashion. Specifically, the GDI-like solubilising factor (GSFs), UNC119 is shown to impact SFK localization in a myristoylation-dependent fashion. SFKs are subsequently released onto recycling endosomal membranes for delivery to the PM by vesicle maturation.

Overall, the manuscript is expected to have an impact on our understanding of the roles of trafficking and chaperones in controlling membrane localization of signaling proteins. Indeed, the roles of differential lipidation and proteins like UNC119 (GSFs, i.e. PDE δ) in regulating peripheral protein localization may represent a general paradigm in cell biology. However, the data are difficult to interpret, as presented. The summary bar plots and histograms are consistent with the interpretation, but do not match the raw data shown. The Src data, in particular, are not very compelling. The manuscript also suffers from being difficult to read and very terse.

Overall, while I agree this is topically important and the author's general line of thinking is highly innovative, I feel this manuscript fails to convince the reader. A number of specific comments are listed below. I do think that a rigorous rewrite along with additional and refined data could significantly improve this paper.

Major comments:

1. Weak plasma membrane localization of Src was observed (Figure 1a and beyond). In contrast to Fyn localization and prior Ras imaging (Schmick, et al., Bastiaens, Cell, 2014), the signal observed is insufficient to argue plasma membrane enrichment of Src. In the Schmick, et al. work, fold enrichment at the PM is clearly quantified; a similar strategy could be extended here to quantify the Src and Fyn localization on the PM and other internal membranes of interest.

2. Ectopic Src expression appears to affect PM localization of other proteins (i.e. Ras, Figure 1a). Exogenous Src, thus, must be demonstrated to not impact cellular phenotype. The fold overexpression of exogenous Src and Fyn over endogenous should also be quantified.

3. In Fig 2a, the fluorescence lifetime of mCit significantly depends on subcellular location. Which cellular compartments are used to calculate the lifetime and alpha values shown in the histograms (iv and v)? WT SFK and UNC show the strongest interactions inside nucleus. This seems unexpected since Src is not localized in the nucleus. It seems that this strong interaction happens only in the case of WT SFK.

4. On line 128, the authors write that Src-mCit interacted with UNC-mCh in the cytoplasm, but not at the PM and the perinuclear region. However, the moderate interactions were seen in the

perinuclear region, i.e. yellow area around the nucleus. The degree of interaction is comparable to that of the 6Q mutant in Figure 2a ii, which is interpreted as a strong interaction fraction on line 146.

5. The HeLa cells used have significant morphological changes in response to siRNA and knockdown throughout the paper. Cell size and ellipticity are highly variable and argue for pleiotropy in these experiments, which makes interpretation extremely challenging. A key example is in Fig 3b ii where one cell is larger than the two in the NT siRNA panel above.

6. In Fig 4, the PLA results are confounding. A significant fraction of the interactions observed are inside of the nuclear region indicated (see item 4 above). Further, all of the distance distributions in PLA are highly similar; the difference in the random distribution for Src is the only deviation. Why is the random distribution for Src shifted toward shorter distance? The similar random distributions should be acquired for all cases. The Src data were acquired in widefield and the others in confocal, so no meaningful conclusion in shifted features of the distribution can be drawn.

7. In Fig 5, authors showed how Arl2/3 regulates SFK distribution in cells using siRNA knockdown (KD) experiments. These results are described in line 186 – KD of Arl2/3 expression gave rise to the decreased concentration of Src-mCit at the PM and an increased soluble fraction as apparent from the enhanced nuclear fluorescence. The nuclear fluorescence may be an insufficient indicator of the soluble fraction. Nuclear protein concentration could depend on various factors, such as expression level and protein interactions. Do you see the same results if you calculate the ratio of plasma membrane intensity and total intensity? Another good way to check effects of Arl on cellular distribution of Src could be expression of constitutively active Arl, e.g. Q70L Arl2.

8. In Fig 6a, authors showed that palmostatin-M treatment resulted in an endomembrane distribution of fully palmitoylated Fyn. This redistribution was explained by a loss in interaction with UNC119. Palmitoylation may not specifically inhibit interactions between UNC and Fyn. This could be explained by prolonged dwell time of palmitoylated Fyn at the Golgi.

9. Arl2/3 activity is not tested (line 191)

10. Strong language is used in the Results section where it may be inappropriate. For example, Lines 123-133 argue that the G2A Src mutant that cannot interact with UNC shows the interaction is via the myristoyl tail. This is not true; it demonstrates that the myristoyl tail plays a role but it could be via allostery or some other mechanism.

Minor comments:

1. In FLIM data, alpha values (fraction of interacting molecules) are negative with minimal or absent protein-protein interactions. How are negative values interpreted?

2. Figure 3d is not discussed

Reviewers' comments:

Reviewer #1 (Remarks to the Author):

Konitsiotis et al.

The authors describe cycles of solubilization, trapping on perinuclear membrane compartments and vesicular transport that maintain the enrichment SFKs at the HeLa cell membrane which is essential for SFK signaling. Similar papers concerning Ras/Pdedelta/Arl2 were recently published (Schmick et al., 2014). The authors propose that UNC119 solubilizes myristoylated SFKs and enhances SFK diffusion to release SFKs on perinuclear membranes mediated by Arl2/3 GDF activity. Src, which cannot be palmitoylated, is trapped on the recycling endosome (RE) from where it traffics to the PM. Fyn is concentrated on the Golgi where it is palmitoylated by a protein S-acyltransferase, and traffics to the PM using the secretory pathway. UNC119 knock down disrupts SFK localization and signaling.

The authors use state-of-the-art technology (Crispr/Cas9, PLA, FLIM, FLAP and FRET) to quantitatively assay interactions and localization of UNC119 and SFKs in HeLa cells. The precise measurement of τ and α is quite impressive.

However, determination of fluorescence intensities is at times difficult to follow, particularly looking at small insets as shown in fig. 1 or black/white representations throughout all figures.

- 1.** We thank the reviewer for her/his constructive comments and we have acted to resolve many of the issues raised. We have restored colour to most images to help determine the different compartments and proteins involved in each experiment. We have also repeated some of the original experiments with improved resolution to better evaluate the localisation of the proteins.

A concern is that all experiments were carried out only in HeLa cells. Are the experiments specific for tumor cells, and can they be repeated in non-tumor cells like hTert-RPE, IMCD3 or haemopoietic cells (Alam et al.)?

2. We addressed this by repeating the SFK localisation and activity experiments with UNC119 knockdown in the breast epithelium cell line MCF10a and obtained similar results (See figure 3a, 3c and Sup. Fig. 6a). This suggests that the SFK spatial cycle is a conserved mechanism within mammalian cells.

The model proposed by the authors calls for the RE (recycling endosome) to be a carrier trafficking SFKs back to the PM, and they identify REs using Rab11a antibodies (Figure 1).

It is impossible to distinguish between RE and perinuclear membranes in DFigure 1, and it is not possible to see where Rab11a really is located. Rab11a is processed at the ER following prenylation and should be present there as well.

3. We have extended our original study in order to better resolve the RE localisation of Src. Specifically, in Figure 1b we co-express a dominant negative mutant of Rab11^{S25N} which inhibits recycling from the RE and we found that Src was now contained within large perinuclear vesicles that completely overlapped with the Rab11^{S25N} fluorescence, while PM localisation was absent. In contrast, Fyn localisation was unaffected which shows that its localisation is not determined by the recycling endosome. Furthermore, in Figure 1c we performed knockdown experiments of both Rab11a and -b and we obtained a similar phenotype of Src trapping in perinuclear vesicular structures, confirming the role of the RE for maintenance of Src PM localisation.

Abstract line 17: The authors state that “The solubilizing factor UNC119 sequesters mislocalized myristoylated SFKs from the cytoplasm, enhancing their diffusion to effectively release SFKs on perinuclear membranes by localized Arl2/3 activity.” It is unclear why UNC119-solubilized SFKs are “mislocalized”? They should be soluble in the cytoplasm and therefore cannot be mislocalized.

- 4.** We agree with the reviewer and have removed this word from the abstract as it was misleading. We were referring to Src being equilibrated to the endomembrane membrane system, where due to rapid spontaneous release from these membranes UNC119 can bind and solubilise the proteins.

It is also unclear what the authors mean with “perinuclear membranes”? The rough endoplasmic reticulum is a perinuclear membrane. Is this what authors are referring to?

- 5.** We are referring to membranes proximal to the nuclear membrane, which are indeed mostly ER. We have now added new data where we show the extensive colocalisation of Arl3 with Rab11 (Figure 4a), showing that the Arl3 mediated localised release of Src from UNC119 occurs mostly on or near the recycling endosome. Arl-mediated release of Src from UNC119 on ER-membranes in the vicinity of the RE will result in rapid “membrane-hopping” of Src (due to lack of electrostatic interactions) until Src becomes trapped through electrostatic interaction with the negatively-charged phospholipids of the RE (See Schmick et al. 2015).

In the introduction (line 71), the authors state that the release of prenylated cargo on perinuclear membranes is mediated by a small GTPase Arl2 (or ARL3) in a GTP dependent manner. This would require that the ARL2- and ARL3-GEFs are located at the perinuclear membranes, as most ARL2/3 will be present in the inactive GDP form that cannot effectively unload myristoylated cargo from UNC119. The ARL3-GEF has recently been identified as ARL13b for which excellent antibodies are available. ARL13b, which is palmitoylated, is more likely present in the cell membrane.

6. We have performed immunofluorescence experiments to show that Arl13b was indeed expressed in HeLa cells and localised to membranes throughout the cell. Importantly, the protein could also be detected in the cilium (as previously reported) in the few cells in which we could detect a cilium by an acetylated Tubulin antibody. We have added this data to this letter for the perusal by the referee. Thus, it may indeed be the case that Arl13b is functional outside the cilium on perinuclear membranes. However, the perinuclear membrane enrichment of Arl2 and Arl3 (Figure 4a), and the studies that have shown that Arl3 interacts with membranes in the GTP bound state (Kapoor et al. 2015), suggest that at least for Arl3 an active Arl3-GEF localises specifically to the RE which might point at other Arl-GEFs that regulate the guanine nucleotide exchange of Arl3 on the RE. This is by itself an interesting finding that merits further investigation. We however believe that this goes beyond the scope of the paper that describes the mechanism that maintains the PM localisation of Src.

Confocal images of HeLa stained with antibodies against Arl13b and acetylated Tubulin, to highlight cilia. The arrowheads indicate where a cilium is present.

The authors may wish to comment on biosynthesis of SFKs, which occurs most likely in the cytosol on free ribosomes. Cotranslational myristoylation is followed by binding to UNC119A/B to freely diffuse throughout the cell? If so, the location of their GEFs is very important. The GEFs will catalyze GDP/GTP exchange and enable unloading of cargo to the GEF containing membrane. Spontaneous

dissociation (line 60) from the membrane should then not matter as SFK-binding to the membrane can be replenished from the pool of soluble SFK/UNC119.

7. The referee is right in thinking that the novo synthesised SFKs released from ribosomes could also be sequestered by UNC119. These *de novo* synthesised SFKs thereby enter a continuous cycle that maintains their PM localisation against entropic equilibration to all membranes. In support of this, we have added new data based on protein synthesis inhibition by cycloheximide to show that Golgi enrichment of Fyn is continuously maintained at steady state by a spatial cycle.

Other Comments

Line 81. Ref 20 is incorrect as Kobayashi et al only showed interaction with ARL2 which is not based on a hydrophobic pocket.

8. We thank the reviewer for bringing this to our attention. We have corrected the reference.

Line 107. "Src-mCit was enriched at the PM and consistently (97% of the cells, n=38) at the Rab11a positive recycling endosome." This is difficult to see. SRC seems to be cytoplasmic in 1a, same for tk-Ras-mCherry. Color for mCit and mCherry may help to localize the proteins. Fyn-mCit localization to the PM is much clearer.

9. We have substantially expanded these experiments to provide a clearer interpretation of the influence of vesicular transport on the steady state localisation of SFKs. We have performed many new experiments to demonstrate the importance of vesicular transport from the RE in maintenance of Src at the PM (Figure 1a-c and Sup. Fig. 2a-b) and of vesicular transport from the Golgi for maintaining Fyn at the PM (Figure 1d-e and Sup. Fig. 2c). We have taken the reviewers advice and restored

colour to most images to aid in better evaluating the data. Furthermore, we have repeated many of these microscopy experiments with improved resolution and better quantified the colocalization.

Line 129. " Src-mCit interacted with both UNC119A-mCh and UNC119B-mCh in the cytoplasm, but not at the PM and the perinuclear region of the cell (Figure 2a-i,-iv,-v..". According to Fig. 2A, the strongest τ and the smallest α are at the nuclear region (or is the perinuclear membrane?), not in the cytoplasm.

10. We thank the reviewer for this observation, which we have now addressed in the text between lines 140-149. We also provided further single-cell analysis to better evaluate the spatial distribution of the SFK-Unc119 interactions (See figure 2a-vi,-vii and 2b-vi, vii). The large fraction of Src-mCit interacting with UNC119-mCherry in the nucleus is due to the complete solubilisation of the small amount of Src-mCit in the nucleus by the molar excess of UNC119-mCherry there. In the cytoplasm, the fraction of Src interacting with UNC119 is lower due to competitive binding to the vast endomembrane surfaces that are absent in the nucleus.

Lines 151-162: Figure 3a and 3d are never mentioned in the text.

11. We have made significant alterations to Figure 3 and made sure that all panels presented are discussed in the text.

Line 165. "Structural and biophysical data indicates that UNC119 interacts with the small GTPases Arl2 and Arl3 in a GTP-dependent manner." A GEF is necessary to catalyzes GDP/GTP exchange. For ARL3, the GEF is membrane bound, and could well be in the PM. ARL3-GTP generated there could well be the trigger for unloading myristoylated cargo. The authors may want to localize ARL13b in HeLa cells, to determine where cargo is unloaded.

We refer the reviewer to the answer 6 above on Arl13b localisation.

Line 182. "Thus, Arl2/3 activity is localised to the perinuclear region, which can lead to a concentration of myristoylated cargos on membranes in this area by localised release from UNC119." If the ARL3 GEF can be localized to the perinuclear region, this would be acceptable.

We refer the reviewer to the answer 6 above on Arl13b localisation.

Line 221: maybe replace "is gained" by "increases"

12. We have adjusted the text accordingly (line 276).

Figure 1: The overlays are too small. Would it not be better to show b/w figures in color? PM localization of src is not easy to see. Fyn at Golgi is also not convincing.

We refer the reviewer to the answers 3 and 9 above on Src/Fyn localisation.

Fig. 2A. The protein interaction between UNC119 and SFK was determined by FRET-FLIM. In Fig. 2A, why is the shortest average fluorescence life time (strongest interaction) at the nucleus (blue color in τ image) not mentioned in the text?

We refer the referee to the answer 10 above on why the strongest τ and the smallest α are at the nuclear region.

Fig. 3a describes the siRNA knockdown of UNC119ab compared with non-targeting siRNA. The cells look very different in I and ii, why?

13. We have performed further experiments in MCF10a cells and looked at the effect on localisation of endogenous SFKs as well as in ectopic expression of SFK-mCit, and obtained similar results. We did not notice any obvious differences in cell shape. We have therefore provided other example images that clearly show no effect of UNC119 KD on cell shape.

While Src-Cit PM localization is decreased, its concentration at the perinuclear region does not change. One would expect that without UNC119, Src would distribute to the whole endomembrane system, as Fyn-Cit does in Fig. 3b ii.

14. The highest density of endomembranes exist in the perinuclear region, which is also due to the highest volume of the cytoplasm there. This is the reason why it is difficult to observe a change in the concentration of protein there upon UNC119 is KD.

In Fig. 3a ii, the authors show Src and ER marker CalR co-localize. Where is Rab11? Is Rab11 colocalizing with SRC? If it does, UNC119 is not necessary for location at the RE and trafficking of SRC to the PM.

15. Due to the large endomembrane density in the perinuclear region, the expected decrease in Rab11 co-localisation upon UNC119 KD is hard to measure. We therefore specifically measured the increase in co-localisation with the ER as UNC119 KD results in a redistribution of Src to the most abundant endomembrane system. We have also explained our reasoning in more detail in the main text (lines 198-203).

Fig. 4a, FRET-FLIM showed Arl2/3-UNC119 interaction throughout the cytoplasm (text lines 166-169). As ARL2/3 interact with UNC119 in a GTP-dependent manner, this would mean ARL-GTP is distributed all over the cytoplasm and thus release of Src from UNC119 could occur throughout the cytoplasm! This is in contradictory to Fig. 4b-c results showing Arl/UNC119 interaction concentrated in perinuclear region.

16. We agree with the reviewer that this is a contradiction. However, in the FLIM experiment the Arl proteins are ectopically expressed which affects the balance between their membrane bound and cytoplasmic fraction in favour of the latter. Because the IF and PLA experiments evaluate this interaction at endogenous protein levels, we circumvent the problem of possible excessive expression of Arl that affects this interaction. We have therefore removed the FLIM experiment from the manuscript.

Fig. 4c. It is not clear how the randomized puncta distribution (green bars) was generated. It's confusing why the distribution pattern is different between Arl/UNC119 (peak around 20um) and Src/UNC119 (peak around 15um). Because the randomized puncta in Src/UNC119 PLA quantification peak around the nuclear area (15um), these cells are not suitable for analysis, as the data could also be interpreted as Src/UNC119 interaction occurring perfectly at perinuclear region (15um from center, similar to Arl/UNC119). The randomized puncta distribution pattern depends both on the relative place of nucleus within the cells (central vs peripheral) and cell size/shape. Ideally, the authors should use those cells with central nucleus for quantification so that the generated randomized

puncta distribution pattern is not center around nucleus. Also, a Src/UNC119 PLA fluorescent image should be provided for Fig. 4c.

17. We have now addressed many of these issues raised by the referee and also provided a more complete explanation in the method section (lines 733-750) of how the randomised distribution was generated. As the reviewer mentions this now better explains the influence of cell shape and size on the shape of the PLA puncta distributions. Furthermore, we performed the Src/UNC119 PLA experiments with confocal microscopy under the same conditions as for the Arl-UNC119 PLA experiments and analysed the PLA distributions in cells of comparable size. By this we obtained a comparable influence of cell shape and size on the PLA puncta distributions. This is apparent from the comparable random distance distributions for the three measured interactions.

Finally, Fig. 4c legend states “cumulative distribution of distance of PLA

puncta.....” (lines 862-863), but this is not cumulative distribution. If it’s cumulative, the percentage will increase with distance and reach 100% at maximal distance.

18. We thank the reviewer for bringing this to our attention, we have now corrected this in the text.

Figure 5: IntDen? Total integrated fluorescence intensity? The figure is confusing as something appears to be below the panels in Fig. 5. What is the redline going through the pictures?

19. We apologise for the confusion generated from this figure. It is indeed total integrated fluorescence intensity. Pink and blue lines indicate the focus point for the

orthogonal views shown above (for YZ) and to the left (for XY) of the micrographs. We have adapted the figure legend to better explain this.

Fig.5b/Fig.6 interpretation and discussion (lines 218-327) are confusing. As UNC119 is obviously important for Fyn PM localization (Fig.3b), there must be solubilized Fyn-UNC119 in the cells. The authors claimed that solubilized Fyn-UNC119 is directly trapped to Golgi by PAT enzyme but not released by Arl2/3 at perinuclear region. There was no evidence provided in this paper to support this claim. Also, it's unclear why the Fyn-UNC119 shows preference for Golgi PAT but not perinuclear Arl2/3, and whether Fyn need to be released from UNC119 before palmitoylation at Golgi (and if so, what the mechanism?).

20. We thank the reviewer for this important issue that was not well explained in our previous version. The point here is not that Arl mediated release of Fyn from UNC does not occur but is not essential for the concentration of Fyn on the Golgi. To increase the kinetics of trapping via palmitoylation of Fyn at the Golgi, rapid UNC119-mediated diffusion of depalmitoylated Fyn through the cytoplasm and spontaneous release from UNC119 (k_{off}) is essential. Because palmitoylated Fyn cannot rebind to UNC119 (supplemental fig 4) it is effectively concentrated on Golgi membranes. By the Arl KD experiments we show that Arl activity has no influence on this kinetic trapping by palmitoylation. This is because Arl-mediated localised release of Fyn from UNC119 on the RE cannot lead to electrostatic trapping because of the lack of a polybasic stretch on Fyn. Any Arl-mediated discharged Fyn on the RE will therefore quickly dissociate to get resolubilised by Unc119. Because Fyn will spontaneously dissociate from UNC119 it will continuously rebind and 'hop' between perinuclear membranes until it gets trapped and concentrated at the Golgi by palmitoylation. We have now added this important interpretation of our data to the discussion (lines 382-391).

Fig. 7, the authors claimed that released Src is trapped to RE and subsequently transported to PM by vesicular trafficking, but evidence is lacking. To exclude the possibility that released Src is also trapped at other organelle (like ER), the author need use additional appropriate experiments (such as Rab11a RNAi, Transferrin labeling etc) to verify that RE is indeed the site of Src partitioning.

21. We have performed the suggestions of the reviewer to resolve the RE localisation of Src. Specifically, in Figure 1b we co-express a dominant negative mutant of Rab11S25N which inhibits recycling from the RE and found that Src was contained within large perinuclear vesicles that completely overlapped with the Rab11S25N fluorescence, while PM localisation was absent. In stark contrast, Fyn localisation was unaffected. Furthermore, in Figure 1c we performed the Rab11a/b knockdown experiments the reviewer recommended, and obtained a similar result. These experiments clearly show the role of recycling in the RE for maintenance of Src PM localisation.

Reviewer #2 (Remarks to the Author):

The authors describe an original mechanism of protein tyrosine kinases Src and Fyn intracellular trafficking. They provide evidence for spatial cycles of solubilisation, trapping on perinuclear membrane compartments and vesicular transport that maintain the enrichment and activity of expressed fluorescence Src and Fyn molecules at the plasma membrane. They also identify the solubilizing factor UNC119, a previously identified Src interactor and activator, as an important component of this molecular process. Functional evidence for this mechanism is provided by a reduced Src activity upon UNC119 silencing in tumor cells, which correlates with a reduced cellular growth in vitro.

The proposed model is potentially interesting; however there are several concerns about the experimental design and on the role of this process in Src signaling that renders the conclusion less convincing.

Major points

1. Data incorporated in this report relies on Src (or Fyn) constructs fused to a fluorescent sequence at the C-terminus. These fusions are expected to stabilize the enzyme in an open and active form, unless a linker is inserted; therefore, the authors need to address whether these constructs alter Src catalytic activity and/or confirm the proposed model with a regulated Src fusion.

22. We thank the reviewer for her/his constructive comments. The fusion proteins do contain a linker between the SFK and the fluorophore and are in the same vector backbone as in Sandilands et al. 2004. Furthermore, we have now included data to show that both proteins are expressed either at the same or lower levels as endogenous SFKs in HeLa cells (Sup. Fig. 1a). We also provide Western blot data (Sup. Fig. 1b) which indicate that starvation leads to inactive Src-mCit or Fyn-mCit in HeLa cells, while PTP inhibition by pervanadate activates both Src-mCit and Fyn-mCit as measured by enhanced pY419 and decreased pY530 phosphorylation. This shows that both fusion proteins have an intact regulation of their activity. Immunofluorescence data (Sup. Fig. 1c) also show that following pervanadate treatment, activation occurs mainly at the PM. Furthermore, we show that the ATP binding site mutant Src-K298M interacts with UNC119 and exhibits a similar localisation to Src-WT. This indicates that activity of Src does not affect the interaction with UNC119. Finally, we address how the SFK spatial cycle described here are consistent with previous results describing the correlation between UNC119 and Src activity (lines 376-381).

2. Most data incorporated in this report relies on overexpressed SFKs molecules, which might force the observed trafficking mechanism. Can the authors provide experimental evidence with endogenous SFKs?

23. We now provide additional immunofluorescence data on the effect of UNC119 KD on endogenous SFK localisation in MCF10a cells (Fig.3c and Sup. Fig. 6a). The interaction of UNC119 with Arl was studied on endogenous proteins by PLA, as was the effect of UNC119 KD on Src activity in the HT-29 cells.

3. It is not clear how this mechanism is regulated. UNC119 was previously described as a Src activator, which raises the question of the role of Src kinase activity in this trafficking process.

24. We have performed new experiments to address this important question. We show that the ATP binding site mutant Src-K298M still interacts with UNC119 and exhibits a similar localisation to Src-WT. This indicates that the active/inactive state of Src does not affect the interaction with UNC119 (lines 192-195). We also discuss (lines 334-339 and 376-377) and substantiate by a new experiment how the previously reported UNC119-induced Src activation can be explained by UNC119-mediated Src spatial cycles. In this new experiment, we show that UNC119-mCh expression, results in decreased SFK activation in HT-29 cells (Fig. 7a-iii and Sup. Fig. 8b). This clearly demonstrates that UNC119 does not activate SFKs by inhibiting the intra-molecular inhibition of activation. Instead, this indicates that increasing the UNC119-SFK interaction by overexpression of UNC119 tips the balance towards SFK solubilisation resulting in decreased SFK activity. This experiment support our overall model that SFK enrichment at the PM as maintained by spatial cycles controls SFK activation and not the UNC119-SFK interaction *per se*.

4. The authors should address how this trafficking function of UNC119 can fit with its previously reported Src regulatory/signaling function. How Src signaling induced by receptors cope with the proposed mechanism?

We refer the referee to our answer 22 above.

5. Functional data supporting an important role for this molecular process is not convincing. UNC119 may interact with several cargos in tumor cells in addition to SFKs; therefore the cellular effect observed upon UNC119 depletion is not sufficient to conclude to an important role of Src-UNC119 trafficking on Src oncogenicity. Besides, Yes seems to play a more important role on HT29 cell transforming properties than Src; however the authors did not address the role of UNC119 on Yes trafficking.

25. We thank the reviewer for this important comment and we have adjusted the discussion (lines 416-425) to how UNC119 affects the spatial localisation of all SFKs. We specifically chose HT29 cells in our UNC119 KD studies because both Src and Yes are overexpressed in these cells and the importance of each SFK to the tumorigenicity of the cell line has been examined directly in two studies: 1) Staley et al. 1997 describe how the specific knockdown of Src in HT-29 cells results in reduced tumorigenicity both in vitro and in mice xenograph studies and 2) Sancier et al. 2011 describe that Yes expression is significant for maintaining tumorigenic potential of HT-29 cells. In the latter study, Src knockdown was not complete not ruling out that Src activity may still be important for tumorigenicity of these cells. Because Yes is palmitoylated it is subjected to the same spatial cycles as Fyn. This is why the KD of UNC119 results in a lower activity of both overexpressed SFKs and reduced growth of HT-29 cells

Minor points

1. Presented data supporting a role for UNC119 on Src trafficking impacting on Src activity is weak; therefore the title should be corrected accordingly, otherwise more experimental data supporting this notion should be incorporated in the ms.

26. We appreciate the reviewers comment and although we have added to the result (lines 309-312) and discussion sections (lines 416-425) to describe how HT-29 cell tumourigenicity is dependent on SFKs, we have decided to follow the reviewers recommendation and removed the word activity from the title.

2. The authors may include a paragraph on what is already known on SFKs trafficking. This would clarify why they became interested in Fyn and why they observed a different trafficking pathway between these kinases.

27. We have significantly restructured and re-written the manuscript to better develop the aims and results of the manuscript. We were juxtaposing the SFK spatial cycle to the Ras spatial cycle previously described, for that reason we wanted to understand how both polybasic stretch containing and palmitoylated SFKs maintain their localisation at the PM. We thus examined both Src (the analogous protein of KRas) and Fyn (the analogous protein to HRas) to show that this spatial cycle is critical for both types of proteins.

3. To my opinion, pictures shown in Figure 1 are too small to have a clear idea of where Src is distributed.

We refer to the answers 3 and 9 to a similar question of referee one. Please note that we have significantly expanded these experiments to provide clearer evidence for the role of vesicular transport from the RE and the Golgi on the steady state localisation of Src and Fyn, respectively.

4. Does this UNC119-dependent trafficking route is used by Src oncogenic alleles?

28. The SFK spatial cycles will affect the localisation of Src irrespective of its oncogenic state, as long as it is myristoylated. Please refer to the answer 24 to the question above on how Src activity does not affect its interaction with UNC119.

5. It is not clear why the designed Cas9 strategy induces a partial KD of UNC119 proteins. On the same line, it is not clear why they overexpress Fyn, which is not linked to colorectal tumor malignancy.

29. The Cas9 strategy resulted in a mixed population of cells upon Tet-induced expression of sgRNA. Although the cells all express the same levels of Cas9 and sgRNA, the induced double strand breaks are repaired via the NHEJ repair pathway which can result in small nucleotide insertions or deletions that can give rise to a diverse array of mutations, a large proportion of which would generate premature stop codons. Because Fyn-mCit exhibited such a distinct PM localisation, it was ectopically expressed in HT-29 cells to better visualise what happens to SFK PM localisation upon Cas9 mediated KD. We have now added this argument in the text (lines 331-333).

6. The effect of UNC119 KD on Src activity can be explained many ways other than a defect on Src trafficking.

We refer the referee to the answer 25 to a similar question above on the effect of Src and/or Yes activity on tumorigenicity of HT-29 cells.

7. Anti-pTyr418 antibodies recognize all SFKs and this point should be corrected accordingly. On the same line, WB incorporated in Fig 8 is puzzling as HT29 cells express aberrant activity of many SFKs

while the blot shows one major band. Finally, I would suggest to confirm the effect on UNC119 on SFK signaling by measuring the phosphorylation level of another substrate.

30. We have adjusted the text and figures to reflect that this antibody recognizes all SFKs. As mentioned in a response above, Src and Yes are highly expressed in these cells, but expression of Src is significantly higher (see Sancier et al. 2011). Src and Yes also have a similar MW (60 versus 61 kDa) that we cannot resolve with our SDS-PAGE gel. We did analyse the effect of UNC119 KD on a known substrate of Src, namely phospho-tyrosine 925 in FAK (Fig7a).

8. Does UNC119 overexpression augment Src transforming properties?

31. We have performed new experiments to examine this question of the referee (see Figure 7a-iii and Sup. Fig. 8b). We found that UNC119 overexpression results in decreased Src activity by increasing the solubilised fraction. This is very much consistent with our model that spatial cycles maintain PM localisation and thereby the activity of Src. This however is inconsistent with previous models that state that UNC119 is a direct activator of SFKs by inhibiting the intra-molecular inhibition of the proteins.

9. The paper is well written and statistical analysis of the experimental data is sound; however some references on Src trafficking and the role of Src in colorectal cancer are missing. Rather, they authors quoted interesting papers on the role of Src in intestinal homeostasis and adenoma formation.

32. We thank the reviewer for pointing at these omissions. We have now included a more detailed section in the discussion addressing the role of SFKs in colorectal cancer (lines 416-425). We also have extensively re-written and re-structured the manuscript to make it more succinct.

Reviewer #3 (Remarks to the Author):

Reviewer comments

The manuscript by Konitsiotis, et al. extends prior work from the Bastiaens group to quantify mechanisms that control the membrane localization of peripheral membrane proteins. Similar to prior work on Ras, the authors argue that two Src family kinases (SFKs, Src and Fyn) require solubilisation and trafficking mechanisms in order to be enriched at the plasma membrane (PM) in an out-of-equilibrium fashion. Specifically, the GDI-like solubilising factor (GSFs), UNC119 is shown to impact SFK localization in a myristoylation-dependent fashion. SFKs are subsequently released onto recycling endosomal membranes for delivery to the PM by vesicle maturation.

Overall, the manuscript is expected to have an impact on our understanding of the roles of trafficking and chaperones in controlling membrane localization of signaling proteins. Indeed, the roles of differential lipidation and proteins like UNC119 (GSFs, i.e. PDE δ) in regulating peripheral protein localization may represent a general paradigm in cell biology. However, the data are difficult to interpret, as presented. The summary bar plots and histograms are consistent with the interpretation, but do not match the raw data shown. The Src data, in particular, are not very compelling. The manuscript also suffers from being difficult to read and very terse. Overall, while I agree this is topically important and the author's general line of thinking is highly innovative, I feel this manuscript fails to convince the reader. A number of specific comments are listed below. I do think that a rigorous rewrite along with additional and refined data could significantly improve this paper.

33. We thank the reviewer for her/his constructive comments and have performed many new experiments and rewritten major parts of the manuscript, which we believe have substantially improved the quality of the manuscript.

Major comments:

1. Weak plasma membrane localization of Src was observed (Figure 1a and beyond). In contrast to Fyn localization and prior Ras imaging (Schmick, et al., Bastiaens, Cell, 2014), the signal observed is insufficient to argue plasma membrane enrichment of Src. In the Schmick, et al. work, fold enrichment at the PM is clearly quantified; a similar strategy could be extended here to quantify the Src and Fyn localization on the PM and other internal membranes of interest.

34. We thank the reviewer for this important comment and have performed new experiments with better resolution that provide a better quantification of SFK PM localisation as well as better address the influence of vesicular transport on the steady state localisation of SFKs. We have substantially extended our study in order to investigate the role of the RE in the localisation of Src. Specifically, in Figure 1b we show how the co-expression of a dominant negative mutant of Rab11S25N that inhibits recycling from the RE, localizes Src onto large perinuclear vesicles that completely overlapped with the Rab11S25N fluorescence, while PM localisation was absent. Important, this Rab11S25N mutant did not affect Fyn localisation. Furthermore, in Figure 1c we show that knockdown of both Rab11a and Rab11b gives rise to a similar accumulation of Src on perinuclear vesicles. To shift the balance in the steady-state Fyn localisation on the Golgi we performed temperature block experiments to slow anterograde transport of proteins from the Golgi to the PM. This allowed a clear identification of the Golgi as a trapping compartment for Fyn (Figure 1d-e and Sup. Fig. 2c).

2. Ectopic Src expression appears to affect PM localization of other proteins (i.e. Ras, Figure 1a). Exogenous Src, thus, must be demonstrated to not impact cellular phenotype. The fold overexpression of exogenous Src and Fyn over endogenous should also be quantified.

35. In our revised figure 1 and the corresponding supplemental figure 2 we now show many examples that ectopic Src expression does not affect the localisation of truncated c-terminal HVR of KRas (tk-Ras). We additionally have included data to show that both fusion proteins are expressed at the same or lower levels than endogenous SFKs in HeLa cells (Sup. Fig. 1a). We also provide Western blot data (Sup. Fig. 1b) that both fusion proteins maintain the regulation of their activity. Immunofluorescence data (Sup. Fig. 1c) also show that following PTP inhibition by pervanadate treatment, SFK activation occurs mainly at the PM. Furthermore, we show that the ATP binding site mutant Src-K298M interacts with UNC119 and exhibits a similar localisation to Src-WT. This indicates that Src activity does not affect the interaction with UNC119.

3. In Fig 2a, the fluorescence lifetime of mCit significantly depends on subcellular location. Which cellular compartments are used to calculate the lifetime and alpha values shown in the histograms (iv and v)? WT SFK and UNC show the strongest interactions inside nucleus. This seems unexpected since Src is not localized in the nucleus. It seems that this strong interaction happens only in the case of WT SFK.

36. We thank the reviewer for her/his careful observations. We have now addressed this issue in the text between lines 140-149 and have provided better quantitative analysis on the spatial distribution of UNC119-SFK interactions (See figure 2a-vi,-vii and 2b-vi, vii). The large fraction of Src-mCit interacting with UNC119-mCherry in the nucleus is due to the complete solubilisation of the small amount of

Src-mCit in the nucleus by the molar excess of UNC119-mCherry. In the cytoplasm, the fraction of Src interacting with UNC119 is lower due to competitive binding to the vast endomembrane surfaces that are absent in the nucleus.

4. On line 128, the authors write that Src-mCit interacted with UNC-mCh in the cytoplasm, but not at the PM and the perinuclear region. However, the moderate interactions were seen in the perinuclear region, i.e. yellow area around the nucleus. The degree of interaction is comparable to that of the 6Q mutant in Figure 2a ii, which is interpreted as a strong interaction fraction on line 146.

37. Similar to answer 36 provided above we have been more specific in the language in that section and described the result in more detail between lines 153-157 and provided better analysis of the spatial distributions for each condition.

5. The HeLa cells used have significant morphological changes in response to siRNA and knockdown throughout the paper. Cell size and ellipticity are highly variable and argue for pleiotropy in these experiments, which makes interpretation extremely challenging. A key example is in Fig 3b ii where one cell is larger than the two in the NT siRNA panel above.

38. We have performed further experiments in MCF10a cells and looked at the effect of UNC119 on localisation of endogenous SFKs as well as ectopically expressed SFK-mCit, and obtained similar results. We did not notice any obvious differences in cell shape. We have now provided more example images to show that UNC119 KD has no effect on cell shape.

6. In Fig 4, the PLA results are confounding. A significant fraction of the interactions observed are inside of the nuclear region indicated (see item 4 above). Further, all of the distance distributions in PLA are highly similar; the difference in the random distribution for Src is the only deviation. Why is

the random distribution for Src shifted toward shorter distance? The similar random distributions should be acquired for all cases. The Src data were acquired in widefield and the others in confocal, so no meaningful conclusion in shifted features of the distribution can be drawn.

39. We thank the reviewer for these insightful points and we have addressed many of the issues. We have provided a more complete explanation in the method section (lines 734-750) of how the randomised distribution was generated. As the reviewer mentions this now better explains the influence of cell shape and size on the shape of the PLA puncta distributions. Furthermore, we performed the Src/UNC119 PLA experiments with confocal microscopy under the same conditions as for the Arl-UNC119 PLA experiments and analysed the PLA distributions in cells of comparable size. By this we obtained a comparable influence of cell shape and size on the PLA puncta distributions. This is apparent from the comparable random distance distributions for the three measured interactions.

7. In Fig 5, authors showed how Arl2/3 regulates SFK distribution in cells using siRNA knockdown (KD) experiments. These results are described in line 186 – KD of Arl2/3 expression gave rise to the decreased concentration of Src-mCit at the PM and an increased soluble fraction as apparent from the enhanced nuclear fluorescence. The nuclear fluorescence may be an insufficient indicator of the soluble fraction. Nuclear protein concentration could depend on various factors, such as expression level and protein interactions. Do you see the same results if you calculate the ratio of plasma membrane intensity and total intensity? Another good way to check effects of Arl on cellular distribution of Src could be expression of constitutively active Arl, e.g. Q70L Arl2.

40. We have now investigated the effect of Arl2/3 KD on Src and Fyn PM localisation in these cells (See figure 5c). As the reviewer correctly notes, a clear decrease in PM localisation is seen in these cells, which significantly adds to the value of these experiments. This along with the results described in figure 6 now better supports the importance of the spatial cycle in maintaining Src localisation. The suggested dominant negative effect of Q70L Arl2 is expected to be very small because the endogenous Arl WT GTPase cycle will still run and maintain Src localisation. We have indeed performed this experiment for Ras cycles (Schmick 2014) with sophisticated image technology and analysis methods. We however do not think that these difficult experiments further add additional information to the Arl KD experiments we show.

8. In Fig 6a, authors showed that palmostatin-M treatment resulted in an endomembrane distribution of fully palmitoylated Fyn. This redistribution was explained by a loss in interaction with UNC119. Palmitoylation may not specifically inhibit interactions between UNC and Fyn. This could be explained by prolonged dwell time of palmitoylated Fyn at the Golgi.

41. This is a valid point to analyse. Because UNC119 sequesters Fyn in the cytosol it does not affect the off rate of lipidated Fyn from membranes. It does however affect the on-rate by increasing the dwell time of Fyn in the cytosol. This results in a net solubilisation of solely Myristoylated Fyn upon UNC expression (Fig2b). Palmitoylation of Fyn decreases the off rate from membranes as compared to solely myristoylated Fyn which indeed decreases the overall amount of UNC119-solubilised palmitoylated Fyn. However, there still should be an increase in solubilisation of fully palmitoylated Fyn upon ectopic UNC119 expression if the proteins interact, which we do not observe (Sup. Fig. 4). In fact, we show that fully palmitoylated Fyn doesn't bind at all to UNC119. We therefore conclude that UNC119 cannot interact with palmitoylated Fyn. We have added part of this argument in the results section (lines 175-178).

9. Arl2/3 activity is not tested (line 191)

42. We have corrected this by replacing 'activity' with 'mediated-release' (line 255).

10. Strong language is used in the Results section where it may be inappropriate. For example, Lines 123-133 argue that the G2A Src mutant that cannot interact with UNC shows the interaction is via the myristoyl tail. This is not true; it demonstrates that the myristoyl tail plays a role but it could be via allostery or some other mechanism.

43. We have significantly rewritten the results and discussion sections where we have been more careful with our statements using wording like 'indicate' where the interpretation is ambiguous. We did however perform more experiments and restructured the paper to clearly show where unambiguous interpretation is valid. In the example that the reviewer mentioned above we have now linked the effect on UNC119-SFK interaction of G2A mutation of SFKs with the mutation in UNC119 binding pocket to come to an unambiguous conclusion.

Minor comments:

1. In FLIM data, alpha values (fraction of interacting molecules) are negative with minimal or absent protein-protein interactions. How are negative values interpreted?

44. These values are not significantly different from zero, which we have now better indicated in Figure 2.

2. Figure 3d is not discussed

45. We have made significant alterations to Figure 3 and made sure that all panels are discussed in the text.

Reviewers' comments:

Reviewer #1 (Remarks to the Author):

the authors have addressed all my concerns satisfactorily.
Publish as is.

Reviewer #2 (Remarks to the Author):

The quality and the clarity of revised ms has been clearly improved and the authors have addressed most of my concerns. In principle I would recommend this work to be published in Nature Communications; nevertheless I would suggest to address to following minor points before publication:

-Summary: To my opinion the authors did not provide solid results supporting the notion that UNC119 can be a drug target; therefore I would recommend to delete this comment in the summary

-Regulation of oncogenic Src by Unc119: To my opinion, the authors did not clearly address this point. The fact that the Src K- close conformation is regulated by Unc119 does not predict that opened conformations of oncogenic Src alleles, such as SrcYF, are regulated by Unc119 in a similar manner. I would be cautious with this conclusion.

-I still have some concerns about the conclusion from Fig7a. It seems that SFK solubilization was performed in a Triton lysis buffer, which does not solubilize SFK localized in lipid rafts (specifically Fyn and Yes); therefore these results could be simply due to a change in SFK membranal repartition. I would suggest to repeat this experiments in lysis conditions that also solubilize SFKs present in cholesterol-enriched domains.

-Finally, It is still not clear to me why the authors addressed Fyn trafficking in HT29 cells and not Yes, which seems to have an important role in transforming properties of these tumor cells. This point is important as the Yes oncogenic role is specified by its SH4 domain (Dubois et al, 2015). What are the evidence for a similar spatial cycle for Yes stated by the authors in point #25 ("...Because Yes is palmitoylated it is subjected to the same spatial cycles as Fyn")? Yes contains a single palmitoylation site unlike Fyn and previous published data suggest a distinct route for Yes trafficking.

Reviewer #3 (Remarks to the Author):

I have gone through the author's detailed response to my criticisms of the original manuscript, as well as those of the other referees. In my opinion, the authors have addressed most of my concerns and I now support publication.

Reviewer #2 (Remarks to the Author):

The quality and the clarity of revised ms has been clearly improved and the authors have addressed most of my concerns. In principle I would recommend this work to be published in Nature Communications; nevertheless I would suggest to address to following minor points before publication:

-Summary: To my opinion the authors did not provide solid results supporting the notion that UNC119 can be a drug target; therefore I would recommend to delete this comment in the summary

We agree with the reviewer that the statement 'qualifying UNC119 as a drug target' is too strong. However, in the manuscript we clearly demonstrate that UNC119 knock down results in reduced phosphorylation of SFKs and associated downstream partner proteins such as FAK. We also demonstrate in the HT29 cancer cells, where Src is a major driving oncogene, that UNC119 KD reduces cell proliferation and increases cell death. We therefore believe that it is not unreasonable to suggest UNC119 as a potential drug target in cancer. We have therefore used a more reserved wording in the summary stating that "Interference with these spatial cycles by UNC119 knock down disrupts SFK localisation and signalling activity, indicating that UNC119 could be a drug target to affect oncogenic SFK signalling".

-Regulation of oncogenic Src by Unc119: To my opinion, the authors did not clearly address this point. The fact that the Src K- close conformation is regulated by Unc119 does not predict that opened conformations of oncogenic Src alleles, such as SrcYF, are regulated by Unc119 in a similar manner. I would be cautious with this conclusion.

The referee expresses remaining concerns that UNC119 could directly regulate Src activity or has selectivity for an active (open) state which stabilizes the active conformation. In other words, the effect of UNC119 knock down on Src activity would not come about by relocalizing Src but rather by inhibiting a hypothetical UNC119-Src interaction-mediated active conformation of Src. We provide evidence in the paper that this cannot be the case. In fig.7a(iii) we show that UNC119 overexpression reduces the activity of SFKs by enhanced solubilization of SFKs (thereby removing SFKs from membranes). If UNC119 would mediate or stabilize an active conformation we should have observed the opposite effect of increased activity of SFKs upon UNC119 expression. This argument was already made in the manuscript in the result section (lines 336-341) and discussion section (378-380) but we have now also added a further sentence in the discussion to this point (381-385). A model of direct UNC119 induced SFK activation is also inconsistent with active Src being at the PM where there is no interaction with UNC119 (Fig.2a). If indeed UNC119 would directly activate Src, its activity should emanate from the cytoplasm, which is clearly not the case.

-I still have some concerns about the conclusion from Fig7a. It seems that SFK solubilization was performed in a Triton lysis buffer, which does not solubilize SFK localized in lipid rafts (specifically

Fyn and Yes); therefore these results could be simply due to a change in SFK membrane repartition. I would suggest to repeat this experiments in lysis conditions that also solubilize SFKs present in cholesterol-enriched domains.

We respectfully disagree with this assessment of the referee that UNC119 could regulate SFK membrane repartitioning. Only a minority of 15-20% of palmitoylated SFKs were observed to partition in these detergent-resistant membrane (DRM) fractions in HT29 cells. This minor fraction contains inactive non-phosphorylated Src kinases and the majority of phosphorylated SFKs and binding proteins are in the detergent soluble fractions (Dubois et al. 2015; PMID: PMC4529617). More importantly, our experiments show that UNC119 knock down reduces the ratio of SFK phosphorylation over total SFK in the detergent soluble membranes. This shows that the *specific* activity of the majority of SFKs that were in the detergent soluble membranes is affected by UNC119 knock down. If UNC119 knock down would repartition SFKs to DRM we would not observe any effect on the specific activity of SFKs. Instead, we should have observed a decrease in the total amount of triton solubilized SFKs and no effect on SFK phosphorylation (because we are blind to the DRM). We instead observed a clear effect on phosphorylation and no effect on triton soluble SFK amounts (total SFK normalized to tubulin is unaltered, for UNC119A sgRNA: from 1.08+/-0.15 before dox induction to 1.05+/- 0.08 after dox induced UNC119A knockdown; for UNC119B sgRNA: from 1.13+/- 0.16 before UNC119 knock down to 0.97+/- 0.2 upon dox-induced UNC119B knock down).

-Finally, It is still not clear to me why the authors addressed Fyn trafficking in HT29 cells and not Yes, which seems to have an important role in transforming properties of these tumor cells. This point is important as the Yes oncogenic role is specified by its SH4 domain (Dubois et al, 2015). What are the evidence for a similar spatial cycle for Yes stated by the authors in point #25 (“...Because Yes is palmitoylated it is subjected to the same spatial cycles as Fyn”)? Yes contains a single palmitoylation site unlike Fyn and previous published data suggest a distinct route for Yes trafficking.

We demonstrated that the localization of both polybasic stretch containing Src and palmitoylated Fyn is dependent on UNC119 and thereby the universality of UNC119-mediated spatial cycles maintaining the localization of these two types of SFKs. This conclusion is supported by our immunofluorescence experiments in MCF10a cells that express polybasic stretch-containing Src, and palmitoylated Fyn and Yes (Kim and Gumbiner, 2015; DOI: 10.1083/jcb.201501025), where the plasma membrane localization of all SFKs was affected by UNC119 knock down. We therefore used Fyn as a SFK localization tracer in the UNC119 knock down experiments in HT29 cells because the change in membrane localization could be better quantified due to Fyn’s clearer PM localization originating from its dual palmitoylation. This was explained in the text (lines 333-334). Furthermore, the observed monopalmitoylated Yes steady-state localization presented in (Sato et al. 2009; doi: 10.1242/jcs.034843), is shifted more to the Golgi than for Fyn, which is consistent with a Yes dynamic palmitoylation cycle analogous to that of Fyn.

REVIEWERS' COMMENTS:

Reviewer #2 (Remarks to the Author):

I would recommend this work to be published in Nature Communications